# The temporal representation of experience in subjective mood

**Hanna Keren[1]\*, Charles Zheng[2], David C Jangraw[3], Katharine Chang[1], Aria Vitale[1], Robb B Rutledge[4,5,6], Francisco Pereira[2], Dylan M Nielson[1†], Argyris Stringaris[1†]**

[1]Section of Clinical and Computational Psychiatry, National Institute of Mental Health, National Institutes of Health, Bethesda, United States; [2]Machine Learning Team, National Institute of Mental Health, National Institutes of Health, Bethesda, United States; [3]Emotion and Development Branch, National Institute of Mental Health, National Institutes of Health, Bethesda, United States; [4]Department of Psychology, Yale University, New Haven, United States; [5]Max Planck UCL Centre for Computational Psychiatry and Ageing Research, University College London, London, United Kingdom; [6]Wellcome Centre for Human Neuroimaging, University College London, London, United Kingdom

**Abstract** Humans refer to their mood state regularly in day-to-day as well as clinical interactions. Theoretical accounts suggest that when reporting on our mood we integrate over the history of our experiences; yet, the temporal structure of this integration remains unexamined. Here, we use a computational approach to quantitatively answer this question and show that early events exert a stronger influence on reported mood (a primacy weighting) compared to recent events. We show that a Primacy model accounts better for mood reports compared to a range of alternative temporal representations across random, consistent, or dynamic reward environments, different age groups, and in both healthy and depressed participants. Moreover, we find evidence for neural encoding of the Primacy, but not the Recency, model in frontal brain regions related to mood regulation. These findings hold implications for the timing of events in experimental or clinical settings and suggest new directions for individualized mood interventions.

**\*For correspondence:**
Hanna.keren@nih.gov

[†]These authors contributed equally to this work

**Competing interests:** The authors declare that no competing interests exist.

## Introduction

Self-reports of momentary mood carry broad implications, yet their underpinnings are poorly understood. We report on our momentary mood to convey to others an impression of our well-being in everyday life (*Clark and Watson, 1988*; *Forgas et al., 1984*); clinically, self-reports of momentary mood form a cornerstone of psychiatric interviewing (*Daviss et al., 2006*; *Wood et al., 1995*); in research, momentary mood is widely used to quantify human emotional responses, such as in ecological momentary assessment (EMA) (*Kahneman et al., 2004*; *Larson et al., 1980*; *Taquet et al., 2020*). Moreover, theoretical accounts suggest that when we report on our mood we integrate over the history of our experiences with the environment (*Eldar et al., 2016*; *Katsimerou et al., 2014*; *Nettle and Bateson, 2012*; *Rutledge et al., 2014*; *Vinckier et al., 2018*). In this paper, we address the fundamental question of the time pattern of this integration—what is the timing of events, for example, early versus recent—that matter the most for how we report our mood.

The standard account is that momentary mood reporting is predominantly affected by recent reward prediction errors (*Rutledge et al., 2014*) (RPEs, or how much better or worse outcomes were relative to what was expected). Accordingly, the more surprising an event is (operationalized as a positive or negative RPE) and the more recent it is, the more it will affect momentary mood. In modeling terms, the standard account posits that humans apply a recency weighting such that their

most recent experiences override those in the more distant past (e.g., experiences that happened early in the course of a conversation or a game would matter far less). This computational account has several real-life implications. In terms of measurement of momentary mood, a momentary happiness rating would be a good proxy for one's most recent experiences. In terms of clinical interactions (such as during an interview or a treatment session), a person's momentary mood could be lifted by the addition of a positive event.

This standard temporal account is widely applied in models of mood (*Eldar et al., 2016*; *Katsimerou et al., 2014*; *Rutledge et al., 2014*; *Vinckier et al., 2018*), yet is largely unexamined and has not been compared to plausible alternatives. Indeed, at the opposite end of the standard recency model stands a primacy account of momentary mood. According to a primacy model, experiences that occur early in a conversation, a game, or interaction prevail over more recent ones. The intuition for such a model comes from idiomatic expressions such as starting off on the right foot or empirical evidence, which shows that the first instances of an interaction can be highly informative (*Ambady and Rosenthal, 1992*; *Houser and Furler, 2007*). Computationally speaking, early events would be weighted more heavily than recent events, which has several real-world implications. From a measurement perspective, the time scales of momentary mood reporting and of experience would overlap less than in the recency model—the current mood rating would be less of a reflection of the current environment. Moreover, the emphasis on the start of interactions such as interviews or treatment sessions would be much greater.

A computational approach can help us answer this important question as it allows us to make explicit in model terms how humans integrate over their experiences in order to arrive at a self-report of their moods (*Huys et al., 2016*). For this purpose, we developed a novel Primacy model that we pitted against the standard Recency model. We then also examine a host of other plausible models as suggested in disparate literatures about valuation timing (*Kahneman and Tversky, 2000*; *Olsson et al., 2017*).

We examine these different temporal integration models across a range of conditions in order to establish their generalizability.

First, we examine the Primacy and the Recency models in their generalizability across reward environments. To do this, we exploited the flexibility of a standard probabilistic task (*Rutledge et al., 2014*) and adapted it to create different task conditions. (1) A random environment, where there was no consistent trend over time in the direction or value of surprises (RPEs); (2) a structured environment, where events in the form of RPEs were organized in positive and negative blocks; and (3) a structured-adaptive environment, where the intensity of RPEs was enhanced in real time to maximize the influence of task events on mood over time and across individuals. We could not be certain that the fixed stimuli of the structured task would be sufficient to drive large changes in mood in each participant, which might influence the temporal integration of events. Therefore, we developed the adaptive task to compensate for individual and temporal differences in mood response, this by delivering personalized stimuli that increase the likelihood of observing a large variation in mood from each participant (implemented by adding a closed-loop controller into the standard probabilistic task).

Second, we examine the generalizability of the different temporal integration models across age groups given that previous studies have shown important differences in reward processing particularly between adolescent and adult groups (*Braams et al., 2015*; *Casey et al., 2010*; *Heller and Casey, 2016*; *Kayser et al., 2015*; *Somerville et al., 2010*; *Walker et al., 2017*).

Third, we also examine the generalizability of these models across healthy volunteers and depressed participants, given the wealth of evidence that depression is associated with aberrations in reward processing (*Keren et al., 2018*; *Ng et al., 2019*; *Stringaris et al., 2015*; *Whitmer et al., 2012*), which might be also affecting the temporal integration of experiences.

We then examine the generalizability of the performance of the two models across simulated data in a model recovery analysis that protects from model selection biases (*Hastie et al., 2009*; *Wilson and Collins, 2019*).

Additionally, we examine the generalizability of the Primacy model performance in comparison to other variants of the Recency model.

Finally, we compare the neural correlates of key terms of these competing models using whole-brain fMRI. Previous work has shown that the reporting of mood and evoking emotional responses leads to activations in a network of brain areas encompassing the fronto-limbic circuit (*Etkin et al.,*

*2015*; *Etkin et al., 2011*; *Rutledge et al., 2014*). Concordance between computational model parameters and neural activity levels provides evidence that the mechanisms described by the model correspond to the neural processes underlying that behavior. For this reason, we test the correlation of Primacy and Recency model parameters with neural activity measured as blood oxygen-level-dependent (BOLD) signal during fMRI and then directly contrast between the relations of the two models.

## Results

### The Primacy and Recency models of mood

As a first step, we compared the Primacy model versus the Recency model of mood. These were designed to correspond to the general experimental setup that is presented in *Figure 1A* and has been used extensively before to answer questions about mood (*Rutledge et al., 2017*; *Rutledge et al., 2014*). In brief, participants first chose whether to receive a certain amount or to gamble between two values. These values allowed each trial to present to the participant an expectation and an RPE value, where the latter is considered as the difference between the outcome and the expectation values. Subjects were also asked to rate their momentary mood every 2–3 trials by moving a cursor along a scale between unhappy and happy mood. Such mood ratings have been shown before to correspond to the general state of well-being of participants (*Rutledge et al., 2017*); we validated this in our dataset with a significant correlation between baseline mood ratings and participant's depressive symptom scores (with Mood and Feelings Questionnaire [MFQ] measure in adolescent sample: CC = −0.62, p = 2.62e-8, CI = [−0.75,–0.44]; with Center for Epidemiologic Studies Depression [CESD] measure in adult sample: CC = −0.69, p = 7.12e-13, CI = [−0.79,–0.56]) and in strong concordance with the gold standard psychiatric interview (KSADS) in distinguishing between patients with depression and healthy volunteers (where the mean initial mood of healthy was also significantly higher than of depressed, t = −3.36, df = 69, p = 0.0012, Cohen's d effect size = 0.97).

The two principal models, Recency and Primacy, are described in *Figure 1B*. Both models consider a cumulative and discounted impact of the expectation term on mood, as shown in *Figure 1B*, *Equation 1* (*Equations 6–8* in Materials and methods provide the complete formulation of these models). The Recency model represents the standard models applied in computational accounts of mood in such setups. In this recency model, expectation is defined as the average between the two gambling values *in the current trial* (*Figure 1B*, *Equation 2*). By contrast, the Primacy model is our hypothesized account of mood in such setups. In this model, expectation is defined as a *weighted average of all previous outcomes* (*Figure 1B*, *Equation 3*). The critical difference between the two models is illustrated below by presenting the different theoretical scaling curves for the influence that events have on mood across the task. As can be appreciated, the Recency model places an emphasis on the most recent trials. By contrast, the Primacy model emphasizes the early ones. The stronger weight of earlier outcomes in the Primacy model emerges from two separate aspects of the model: first, that one's expectation for the next outcome is based on the average of all previously received outcomes and, moreover, that mood is determined by the sum of all such past expectations (see *Figure 1—figure supplement 1* for a graphical illustration of these two aspects). The Primacy model, therefore, is unique from the prior Recency model because it defines expectation on the basis of the full history of events. We should note though, since the RPEs are recency weighted and are calculated based on the difference of the (primacy weighted) expectation and actual outcome, that recency-weighted outcomes do still influence mood in the Primacy model.

### Primacy versus variants of the Recency model

Next, we tested the Primacy model performance against several alternative variants of the Recency model.

First, we addressed the learning component implemented in the Primacy model by creating a Recency model with learning from previous trials (termed the 'Recency with dynamic win probability model'). This model considered the actual individual winning probability instead of a fixed win probability of 50% in the expectation term (using a trial-level individual winning probability derived from the percentage of previous win-trials).

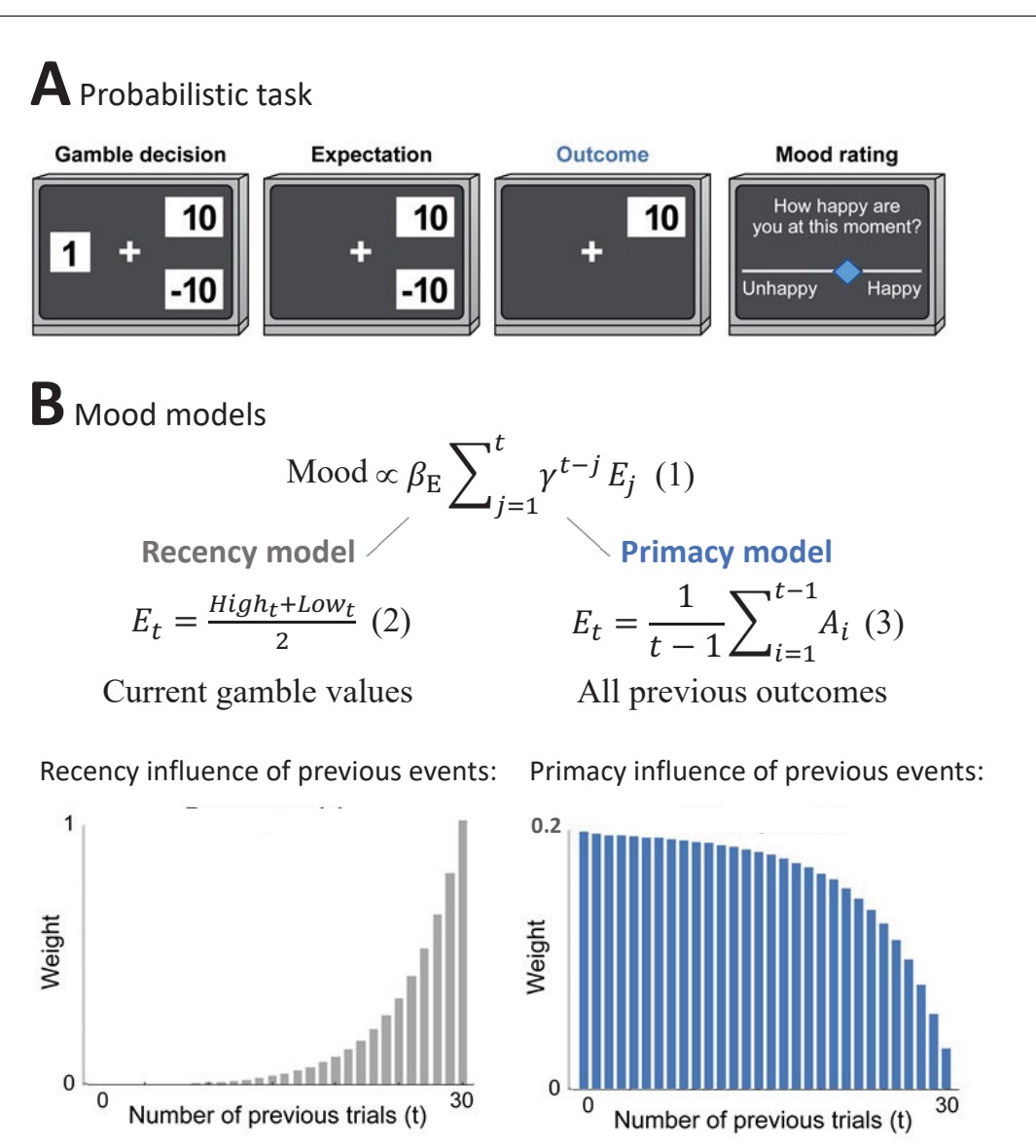

**Figure 1.** The Primacy versus Recency mood models. (**A**) Participants played a probabilistic task where they experienced different reward prediction error values, while reporting subjective mood every 2-3 gambling trials. In each trial, participants chose whether to gamble between two monetary values or to receive a certain amount (Gamble decision). During Expectation, the chosen option remained on the screen, followed by the presentation of the Outcome value. (**B**)

$$\text{Mood} \propto \beta_E \sum_{j=1}^{t} \gamma^{t-j} E_j \qquad (1)$$

presents the expectation term of the mood models, where $\beta_E$ is the influence of expectation values on subjective mood reports. The expectation term of the Recency mood model as developed by *Rutledge et al., 2014* is presented below

$$E_t = \frac{High_t + Low_t}{2} \qquad (2)$$

where it consists of the trial's high and low gamble values. In the alternative Primacy model, as presented in
*Figure 1 continued on next page*

*Figure 1 continued*

$$E_t = \frac{1}{t-1}\sum_{i=1}^{t-1} A_i \tag{3}$$

the expectation term is replaced by the average of all previous outcomes ($A_i$). Moreover, as can be seen in *Equations 6–8* in Materials and methods, the Primacy model has overall fewer parameters compared to the Recency model. The theoretical scaling curves for the influence of previous events on mood due to expected outcomes are presented for each model respectively below (see *Figure 1—figure supplement 1* for additional illustrations).

The online version of this article includes the following figure supplement(s) for figure 1:

**Figure supplement 1.** The Primacy effect of outcomes on mood.

Specifically, the win probability was formulated as follows:

$$p(t) = \frac{\sum_{t=1}^{t} I(A(t-1) = H(t)) + 5}{\sum_{t=1}^{t} G(t-1) + 10} \tag{4}$$

where the sum in the numerator counts the previous trials on which the outcome was the higher value H (I is a binary vector of this condition with 1 for outcome H and 0 for the lower outcome L) and the sum in the denominator counts the previous trials on which the participant chose to gamble (G is a binary vector of this condition with 1 for the choice to gamble and 0 for the choice of the certain value). The additional bias of 5 in the numerator and 10 in the denominator implement Bayesian shrinkage corresponding to 10 prior observations with an average success of 0.5. The expectation term was modified accordingly:

$$E_t = (p(t)H(t) + (1 - p(t))L(t)) \tag{5}$$

We then addressed the elimination of the Certain term in the Primacy model by testing it against a 'Recency without the Certain outcome' model.

Additionally, we addressed the unique feature of the Primacy model where individuals use experienced outcomes to generate their expectations. We therefore compared the Primacy model to a Recency model where the expectation term is based on the previous outcome rather than current trial's gamble values (the 'Recency with outcome as expectation model').

Next, we compared the Primacy model to a Recency model that merges the dynamic win probability and the elimination of the Certain term modifications, which is the most similar Recency model to the Primacy model (termed the 'Recency with both dynamic win and no Certain model').

See *Supplementary file 2* for the formulation of these alternative Recency models.

## Primacy versus Recency models comparison criteria

We started from using two main criteria to compare between the models. First, a training error, the mean squared error (MSE) of fitting the model to participant's mood ratings. Second, a streaming prediction error, a within-subject prediction of each mood rating using the preceding mood ratings (with first 10 mood ratings being discarded as we found that the streaming prediction error criteria were unstable in the first trials due to fewer available data points). A model performed better if it had significantly smaller error between predicted and rated mood values in these criteria, as tested across participants with a one-sided Wilcoxon signed-rank test, with $p < 0.05$ (tests the null hypothesis that two related paired samples come from the same distribution). We chose a one-sided null because the conservative null would be that the new approach is equal to or worse than the existing approach. Moreover, we used a leave-out sample validation and independent confirmatory datasets in all model comparisons.

We then performed a model recovery assessment to validate the model selection criteria by which we compare the performance of the Primacy and Recency models. We first generated simulated datasets using each of the models and then fit the models and tested whether we could correctly identify the model that generated the data. According to both the training error and the streaming prediction criteria, it was possible to recover the true model from the simulated data (the

Recency model performed better on data simulated with the Recency model, and vice versa; see *Supplementary file 1*). We then preferred to use the streaming prediction error across all comparisons as it is a more valid criterion due to the training error favoring overfitting (*Hastie et al., 2009*).

### Primacy versus Recency models across different reward environments

Here, we compare the models and assess their validity across differently structured reward environments.

#### Random environment

In order to generate a random reward environment, we used the standard probabilistic task (*Rutledge et al., 2014*) as described above, where the RPE values were drawn randomly from a pre-defined range of values (*Figure 2A*). As shown in *Figure 2B* (left panel), this causes mood fluctuations in keeping with previous results (presents the mean across n = 60 participants, with a significant effect of linear time, but not squared time, on mood in a linear mixed-effects model: $\beta_{time}$ = -0.31, SE = 0.11, p = 0.006; $\beta_{time^2}$ = 0.0009, SE = 0.002, p>0.05, and mood change effect size [mean ± SD] = -0.93 ± 1.70).

Comparing the Primacy versus the different Recency models, we found that the Primacy model outperformed each of the Recency models on the streaming-prediction criterion (see *Figure 3* and *Table 1* for model performance comparison and *Figure 3—figure supplement 1* for the distributions of the fitting coefficients of each of the tested models in this environment).

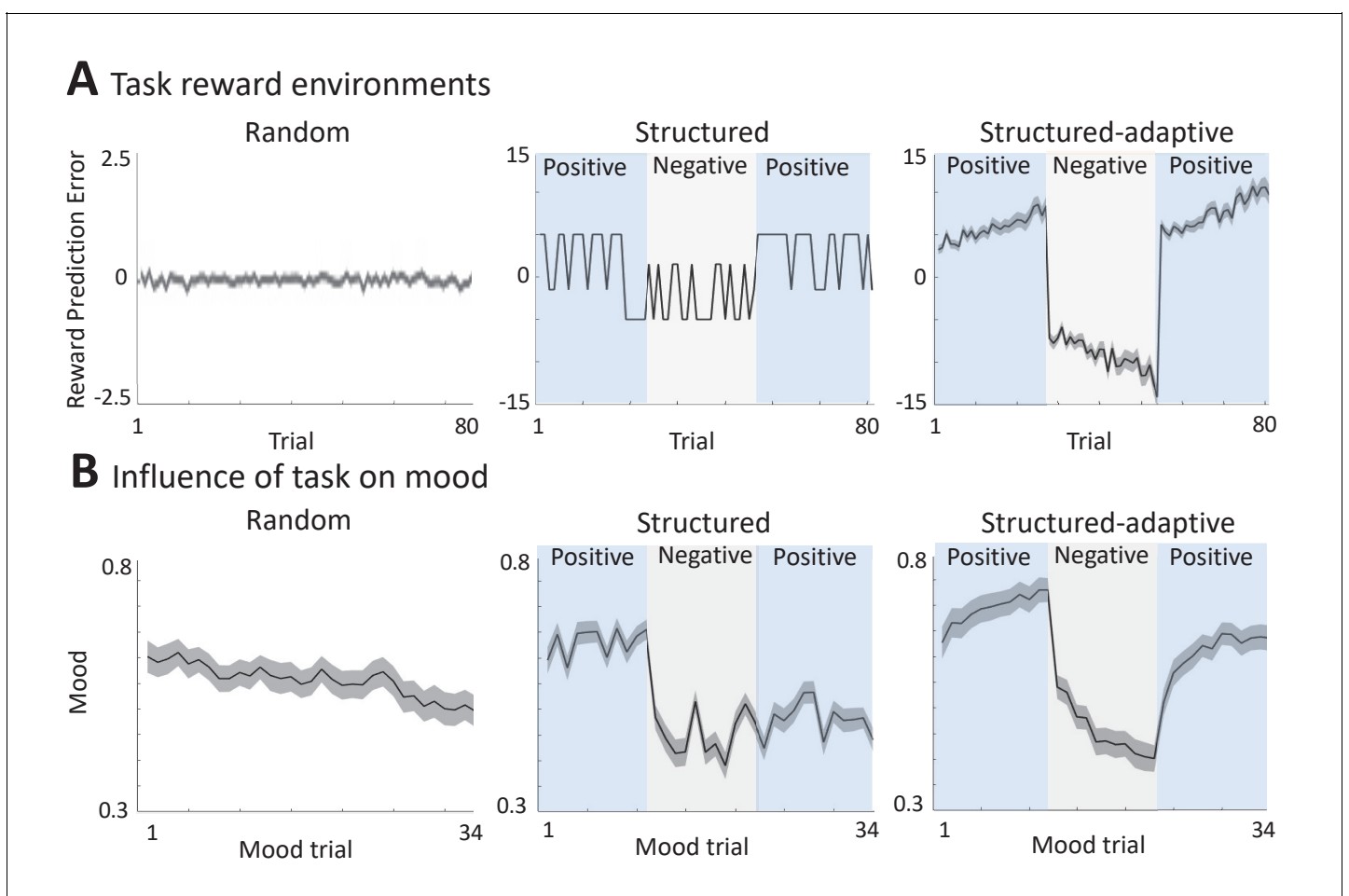

**Figure 2.** Different experimental reward environments. (**A**) Reward prediction error (RPE) values received during each task version, averaged across all participants (shaded areas represent SEM). (**B**) The influence of RPE values on mood reports along the task, averaged across all participants (shaded areas are SEM). See Materials and methods for a link to the online repository from where the source data of this figure can be downloaded.

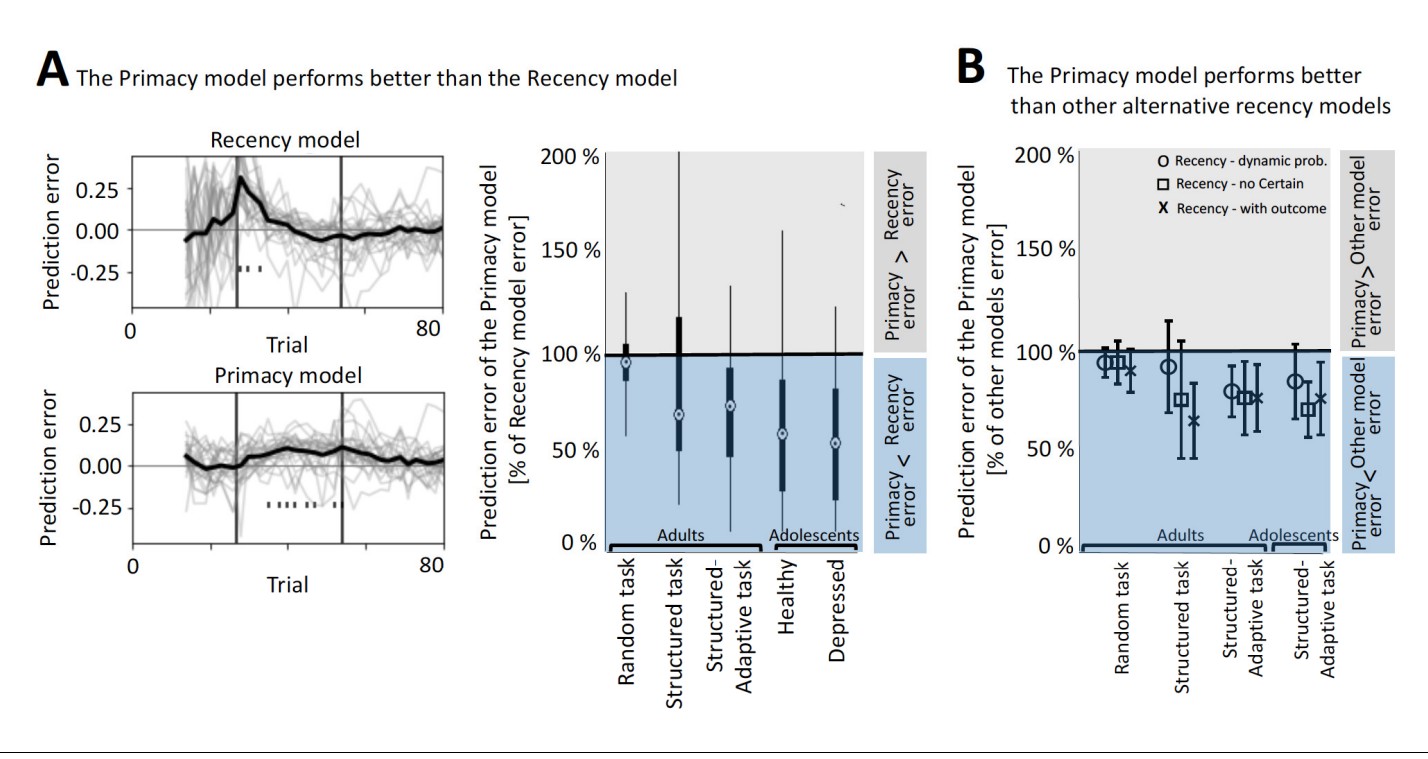

**Figure 3.** The better performance of the Primacy model. (A) Model comparison between the Primacy and the Recency models, using the streaming prediction criterion, where the model is predicting each mood rating using the preceding ratings. On the left, the trial-level errors in predicting mood with the Recency and the Primacy models are shown for all participants, during the structure-adaptive task (bold line depicts average across all participants). This error is calculated by predicting the t-th mood rating using all preceding (1 to t-1) mood ratings, and therefore fitting iterations start only as of the fourth mood rating (~15 gambling trials), which ensures that models have sufficient data to fit all parameters. The right panel presents the median of mean squared errors (MSEs) of the Primacy model relative to the Recency model in this criterion across all datasets (edges indicate 25th and 75th percentiles, and error bars show the most extreme data point not considered an outlier). (B) Model comparison between the Primacy and three variants of the Recency model, which also shows lower MSEs for the Primacy model. Values are median MSEs of the Primacy model relative to each of the alternative models (i.e., Recency with dynamic probability model marked with circles, Recency without the Certain term model marked with squares, and the Recency with outcomes as expectation model marked by crosses), and error bars are standard deviation across participants. The values used to derive these plots are available in *Table 1*, fit coefficients are presented in *Figure 3—figure supplement 2*, and a link for downloading all the modeling scripts can be found in Materials and methods.

The online version of this article includes the following figure supplement(s) for figure 3:

**Figure supplement 1.** Distributions of the estimated coefficients for the parameters of the Primacy and the different Recency models.

**Figure supplement 2.** Expanding the Primacy model.

**Figure supplement 3.** Mood ratings and the respective trial-wise model parameters.

### Structured environment

In order to generate an environment that has some structure (consistently positive or negative events), we modified the probabilistic task as shown in *Figure 2A* (middle panel): RPE values were divided into three blocks, one of positive RPEs (+5), the second of negative RPEs (–5), and a third block of positive RPEs again (+5). We found that the experimental setup leads to substantial fluctuations in mood as can be seen in the middle panel of *Figure 2B* (presents the mean across n = 89 participants, with a significant effect of time, both linear and squared, on mood in a linear mixed-effects model: $\beta_{time}$ = -0.02, SE = 0.006, p<0.0001; $\beta_{time^2}$ = 0.0004, SE = 0.0001, p = 0.009, and an effect size per block of [mean ± SD]: 0.56 ± 1.90, –1.42 ± –1.42, –0.55 ± –0.55, for the first, second, and third blocks, respectively).

Comparing the Primacy versus the different Recency models, we found that the Primacy model outperformed each of the Recency models on the streaming prediction criterion (see *Figure 3* and *Table 1* for model performance comparison and *Figure 3—figure supplement 1* for the distributions of the fitting coefficients of each of the tested models in this environment).

**Table 1.** Performance of the Primacy model versus alternative Recency models in three different reward environments (in adults) and in a lab-based sample comprising adolescent participants, of which 40% were diagnosed as clinically depressed (using the structured-adaptive task).

Statistical comparison is of the streaming prediction errors. (MSE: mean squared error; IQR: interquartile range).

| | | Model | MSE median | IQR | z-Value | p-Value |
|---|---|---|---|---|---|---|
| Reward environment | Random task | Primacy | 0.0165 | 0.0099 | - | - |
| | | Recency | 0.0171 | 0.0091 | 1.8480 | 0.0323 |
| | | Recency with dynamic win probability | 0.0170 | 0.0078 | 2.7440 | 0.0030 |
| | | Recency without a Certain term | 0.0176 | 0.0099 | 1.9973 | 0.0229 |
| | | Recency with outcome as expectation | 0.0187 | 0.0114 | 3.0053 | 0.0013 |
| | | Recency with both dynamic win and no Certain | 0.0171 | 0.0079 | 2.7440 | 0.0030 |
| | Structured task | Primacy | 0.0088 | 0.0036 | - | - |
| | | Recency | 0.0097 | 0.0069 | 1.6613 | 0.0483 |
| | | Recency with dynamic win probability | 0.0090 | 0.0043 | 1.8853 | 0.0297 |
| | | Recency without a Certain term | 0.0109 | 0.0091 | 3.0053 | 0.0013 |
| | | Recency with outcome as expectation | 0.0141 | 0.0044 | 3.8266 | 0.0001 |
| | | Recency with both dynamic win and no Certain | 0.0090 | 0.0044 | 1.8853 | 0.0290 |
| | Structured adaptive | Primacy | 0.0137 | 0.0041 | - | - |
| | | Recency | 0.0160 | 0.0040 | 3.4533 | 0.0003 |
| | | Recency with dynamic win probability | 0.0171 | 0.0040 | 3.7146 | 0.0001 |
| | | Recency without a Certain term | 0.0189 | 0.0060 | 3.5279 | 0.0002 |
| | | Recency with outcome as expectation | 0.0179 | 0.0063 | 3.6773 | 0.0001 |
| | | Recency with both dynamic win and no Certain | 0.0172 | 0.0040 | 3.6770 | 0.0001 |
| Age | Adolescents lab-based | Primacy | 0.0066 | 0.0021 | - | - |
| | | Recency | 0.0077 | 0.0028 | 3.4533 | 0.0003 |
| | | Recency with dynamic win probability | 0.0079 | 0.0026 | 2.8559 | 0.0021 |
| | | Recency without a Certain term | 0.0094 | 0.0029 | 3.9013 | 0.0000 |
| | | Recency with outcome as expectation | 0.0093 | 0.0038 | 3.6773 | 0.0001 |
| | | Recency with both dynamic win and no Certain | 0.0079 | 0.0027 | 2.9306 | 0.0017 |
| Diagnosis | Depressed adolescents | Primacy | 0.0043 | 0.0069 | - | - |
| | | Recency | 0.0072 | 0.0053 | 3.2666 | 0.0005 |
| | | Recency with dynamic win probability | 0.0075 | 0.0042 | 3.3039 | 0.0004 |
| | | Recency without a Certain term | 0.0074 | 0.0042 | 3.3786 | 0.0003 |
| | | Recency with outcome as expectation | 0.0089 | 0.0043 | 3.9013 | 0.0000 |
| | | Recency with both dynamic win and no Certain | 0.0086 | 0.0069 | 3.4159 | 0.0003 |

## Structured-adaptive environment

Since there can be substantial individual differences in response to events in the environment, we developed a third, adaptive task that tracks individual performance and modifies the environment accordingly. In this paradigm, RPE values were not predefined but modified in real time and in an individualized manner by a proportional-integral (PI) controller (*Levine, 2011*) to enhance their potential positive or negative influence on mood over time (see rightmost panel of *Figure 2A* for the average RPE values across all n = 80 participants). This task also consisted of three blocks of RPE values, pushing mood towards the highest mood value in the first block, the lowest mood in the second, and the highest mood again in the third block. We found that this experimental setup leads to the largest changes in mood as can be seen in the rightmost panel of *Figure 2B* (a significant effect of time, both linear and squared, on mood in a linear mixed-effects model: $\beta_{time}$ = -0.04, SE = 0.004,

p<0.0001; $\beta_{time^2}$ = 0.001, SE = 0.0001, p<0.0001, and an effect size per block of [mean ± SD]: 0.92 ± 1.60, –1.75 ± 1.10, 1.45 ± 1.70, for the first, second, and third blocks, respectively).

Comparing the Primacy versus the different Recency models, we found that the Primacy model outperformed each of the Recency models on the streaming prediction criterion (see *Figure 3* and *Table 1* for model performance comparison and *Figure 3—figure supplement 1* for the distributions of the fitting coefficients of each of the tested models in this environment).

In what follows, we tested the performance of the Primacy model versus the Recency models, also across different age groups (including different experimental conditions), and across depressed participants.

## Primacy versus Recency models across different age groups

We found no differences in the strength of the mood changes by age group (no significant group effect on mood in a linear mixed-effects model: F(152,1) = 2.35, p = 0.12, between an adult sample [n = 80] with mean age ± SD = 37.76 ± 11.23, versus an adolescent sample [n = 72] with mean age of 15.49 ± 1.48). The collection of these two datasets differed in age but also in experimental conditions as the adult sample was collected online (see Materials and methods for details and the pre-registered analysis link), while the adolescent sample was a lab-based collection in an fMRI scanner.

We found that the Primacy model outperformed the different Recency models in both the online adult and the lab-based adolescent samples (see *Figure 3* and *Table 1* for model performance comparison and *Figure 3—figure supplement 1* for the distributions of the fitting coefficients in each age group).

## Primacy versus Recency models across different diagnostic groups

We found no differences in the strength of the mood changes between the healthy and depressed adolescent participants (when controlling for the difference in baseline mood, there was no significant group effect on mood in a linear mixed-effects model: F(70,1) = 0.77, p = 0.38; between healthy participants [n = 29] with mean ± SD depression score [MFQ] = 1.84 ± 2.49, versus participants diagnosed with major depression disorder [n = 43 with mean depression score of 8.31 ± 6.27; 12 or higher being the cutoff for indicating depression]).

Comparing the Primacy model versus the different Recency models in the depressed adolescent sample, we found that the Primacy model outperformed each of the Recency models (see *Figure 3* and *Table 1* for model performance comparison). Moreover, we confirmed the superior performance of the Primacy model result also in adult participants with high risk for depression (n = 28 participants with CESD scores above 16, being the cutoff for high risk for depression, showed significantly lower MSEs for the Primacy against the Recency model in a Wilcoxon test with p<0.001).

For completeness, we also tested the Primacy model against models with other weighting of past events. *Figure 3—figure supplement 2* presents a model in which we added to the expectation term a decay parameter and a parameter for how many previous outcomes are included, resulting in various possible scaling curves for the influence of previous events (Equations S1 and S2). Comparing five such alternative models showed that the Primacy model outperformed these models too (significantly lower streaming prediction errors for the Primacy model in a Wilcoxon test with p<0.001).

See also *Figure 3—figure supplement 3* for a demonstration of the change over time in task values and the Primacy versus the Recency model parameters, shown both for a single participant and on average across the group.

## Primacy versus Recency models in relation to brain responses

Finally, we compared the Primacy and the Recency models on the basis of their relationship to brain activity measured using fMRI. To this end, participants were scanned whilst completing the structured-adaptive version of the task. We correlated BOLD signal with the participant-level weights of the parameters of the Primacy and two of the Recency models (the original Recency model and the Recency model that is most similar to the Primacy model, i.e., the one with both dynamic win probability and no Certain term). We found that neural activity preceding the mood rating phase (*Figure 4A*) was significantly correlated to the Primacy model expectation term ($\beta_E$), which reflects the relationship between mood and previous events (*Figure 4B*, cluster at the anterior cingulate cortex [ACC] and ventromedial prefrontal cortex [vmPFC] regions, n = 56, peak beta = 44.80, t = 3.37

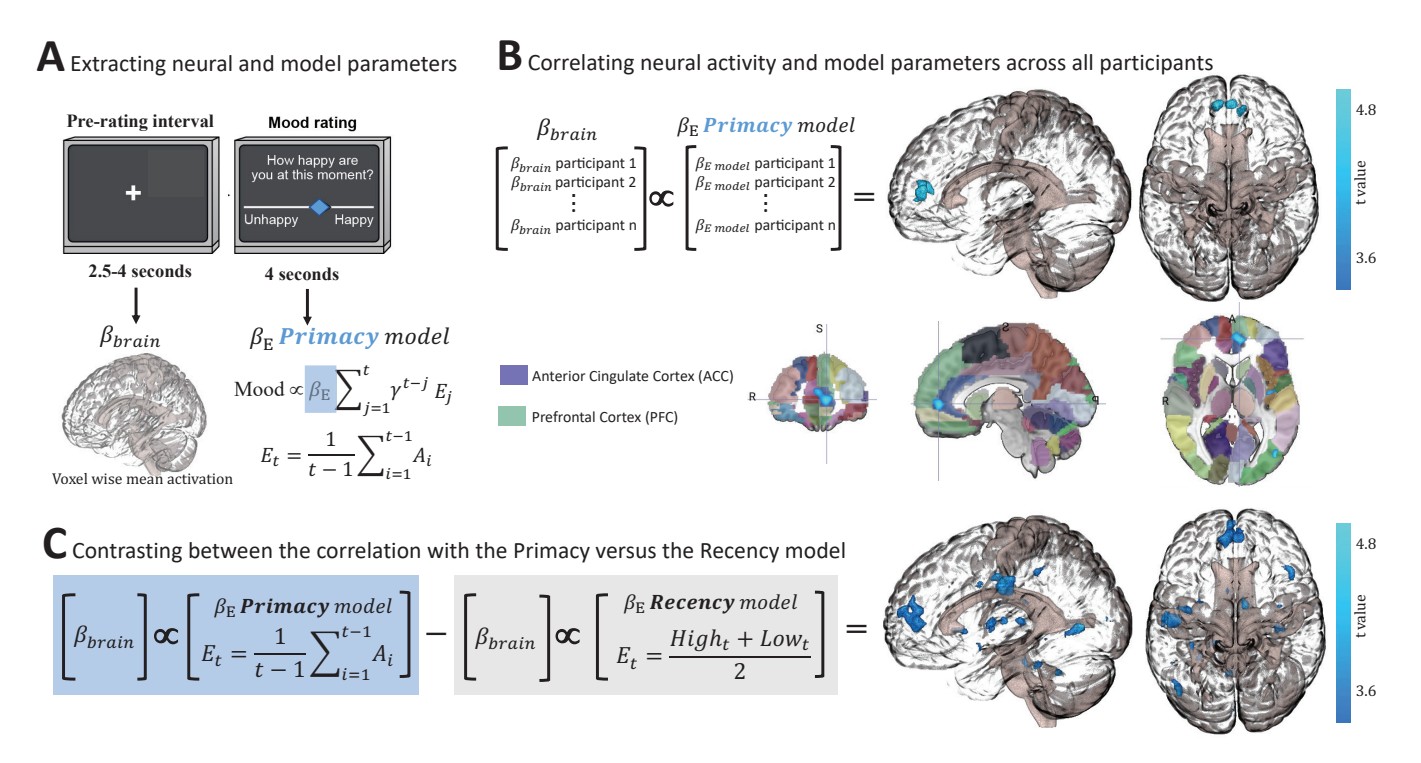

**Figure 4.** Neural correlates of the Primacy model. (**A**) Extracting individual whole-brain BOLD signal activation maps ($\beta_{brain}$) during the time interval preceding each mood rating, and individual model parameters by fitting mood ratings with the Primacy model ($\beta_E$). (**B**) Correlation across participants between the individual weights of the model expectation term, $\beta_E$, and the individual voxel-wise neural activations. A significant cluster was received with a peak at [–3,52,6], size of 132 voxels, threshold at p = 0.0017 (after a multiple comparisons correction as well as a Bonferroni correction for the three 3dMVM models we tested). Below, the resulting cluster of significant correlation is presented aligned on the Automated Anatomical Labeling (AAL) brain atlas for spatial orientation (focus point of the image is at [–7.17,50,4.19], which is located in the ACC region). (**C**) A statistical comparison between the relation of brain activation to the Primacy versus the Recency models. We compared the regression coefficients of the correlation between participants' brain activation and the Primacy expectation term weights versus the regression coefficients of the relation to the Recency model expectation term (see *Figure 4—figure supplement 1* for the two images before thresholding and before contrasting against each other). This contrast showed a significantly stronger relation of the Primacy model expectation weight to brain signals at [–11,49,9], extending to a cluster of 529 voxels (p = 0.0017). See Materials and methods for a link to the online repository from where the neural analyses scripts and the presented images can be downloaded.

The online version of this article includes the following figure supplement(s) for figure 4:

**Figure supplement 1.** Uncorrected raw data neural correlates of the Primacy model and two Recency models, the original one and the one with the most similar characteristics to the Primacy model (with both dynamic win probability and elimination of the Certain term).

**Figure supplement 2.** Mood encoding at the whole-brain level in the structured-adaptive task: mood encoding values are derived using the mood ratings as the parametric linear modulator of the BOLD signals during the pre-rating interval (at this interval, which lasts between 2.5 and 4 s, participants are presented with the mood question, but cannot rate their mood yet).

and p = 0.0017, which corrects to p<0.05, first by using the 3dClustSim in AFNI software [Analysis of Functional NeuroImages] with an autocorrelation function [ACF] resulting in p = 0.005 and a minimal cluster size of 100 voxels followed with a Bonferroni correction since we compared three different models, reaching p = 0.005/3 = 0.0017). By contrast, both Recency models' individual parameters showed no significant relation to neural activity. To formally compare between the two models in their relationship to brain activity, we contrasted the two voxel-wise correlation images, that is, the BOLD signal correlation across participants with the Primacy model individual weights $\beta_E$ (coefficients of the expectation term) versus BOLD signal correlation with the Recency models $\beta_E$. This showed a significantly stronger relation of the Primacy model to neural activity, specifically in the ACC and vmPFC regions (*Figure 4C*, t = 5.00, using the corrected threshold of p = 0.0017). This result provides a possible neural underpinning specific to the Primacy model's mathematical realization of expectations and mood. Additionally, mood ratings were correlated to the preceding neural

activity level in the striatum during the structured-adaptive task (*Figure 4—figure supplement 1*), in congruence with previous accounts of mood relations to striatal activity in the random task design.

## Discussion

A fundamental assumption about how humans report on their mood is that they integrate over the history of their experiences. In this paper, we sought to test this assumption and establish the temporal structure of this integration.

We show that when humans report on their momentary mood, they do indeed integrate over past events in the environment. However, we find no support for the commonly assumed recency model of integration, that is, for the assumption that most recent events matter the most in this integration. Instead, we find several lines of evidence to support a primacy model of the integration of past experiences in mood reports.

The first line of evidence comes from comparing the primacy, the recency, and several other models in a probabilistic task that has been widely used in the past to influence subjective mood reports. This version of the probabilistic task presents participants with random RPE values (*Rutledge et al., 2014*). The model comparison, conducted using errors of within-subject prospective prediction of consecutive mood ratings, indicated that a primacy model, that is, a model that places greater and long-lasting emphasis on early events, described mood ratings the best. The Primacy model also outperformed several other plausible models, including an extended model that allowed other timing of previous events to be most influential, as well as models resulting from modifications of the Recency model (where terms were excluded or replaced by alternative task values).

We then sought to test whether our findings generalized beyond a random reward environment. We did so in order to emulate real-life situations where negative or positive events tend to cluster over periods of time, such as a conversation between two people that can include consistently pleasant (or unpleasant) events throughout the interaction time frame. We did this by adapting the probabilistic task in two different ways. First, by introducing blocks of predetermined consecutively negative or positive RPEs. In this structured environment, the Primacy model outperformed the alternative Recency models. We also modified the probabilistic task in another way, namely by introducing a PI control algorithm. This created a structured-adaptive task, with a block of consecutively negative or positive RPEs that were, however, tailored in real time to each individual's initial mood level and mood response, to maximize the influence of these events (such as when we modify our tone of speech in real time during a conversation, e.g., according to who we are interacting with and the response we aim for). The Primacy model clearly outperformed the alternative recency models also in this task.

It is conceivable that what appears to be a primacy effect is actually due to longer-lasting effects of, say, positive RPEs on mood—this could be particularly exacerbated in the structured-adaptive task. However, since our result was also robust in a random task design, as well as when testing a model with a varying time window parameter that considered different number of previous trials ($t_{max}$, see *Figure 3—figure supplement 2*), we do not find evidence for the block valence order to account for the better performance of the Primacy model. In addition, the interaction between individual behavior and the controller in the structured-adaptive environment could raise interpretative difficulties. Therefore, it is important that the Primacy model also fit better in the structured and random tasks, where the tasks did not respond to individual differences in responsiveness to RPEs. Yet, it is possible that this is a contribution to the fact that the advantage of the Primacy model over the Recency models is greater in the structured-adaptive task. There may also be additional mechanisms at play in the structured-adaptive task such as hedonic extinction towards RPEs that explain some of the increased performances of the Primacy model compared to the Recency model in this task.

We then also sought to test whether our findings generalized across two important variables. One such variable is age. Substantial evidence shows that adolescence is a time when levels of self-reported mood can change dramatically, for example, through overall increases in the levels of depression (*Heller and Casey, 2016*; *Maciejewski et al., 2015*; *Ronen et al., 2016*; *Stringaris and Goodman, 2009*). Also, adolescence marks a time when reward processing appears to be different to that of adulthood with reported increases in the sensitivity of mood (*Braams et al., 2015*; *Casey et al., 2010*; *Heller and Casey, 2016*; *Kayser et al., 2015*; *Somerville et al., 2010*;

*Walker et al., 2017*). Our primacy model fitted better than the recency alternative in both adolescent and adult samples.

The other variable is subjects' depression. Its importance is twofold. First, the self-evident fact that persons with depression report a lower mood than non-depressed persons, this is the case in clinical but also in the experimental task setting we and others have used (*Rutledge et al., 2017*). This difference in mean scores could be reflecting a different way in which persons with depression report on their mood. In keeping with this, the integration of experiences could be happening in a different temporal structure. The second reason is that persons with depression are thought to display reward processing aberrations (*Keren et al., 2018*; *Ng et al., 2019*; *Stringaris et al., 2015*; *Whitmer et al., 2012*)—for example, in the form of being less sensitive to rewards or learning less from them—that could impact the way in which they integrate across environmental experiences. We address this question specifically in adolescents, the time of a sharp increase in depression incidence (*Beesdo et al., 2009*), and find no evidence that depressed adolescents applied a different model to the one that their non-depressed counterparts did. Moreover, also adult participants with high scores of depression showed that the Primacy model is a better account of their mood reports. These findings strongly suggest that the temporal representation of experiences offered by the model is robust to important personal characteristics.

A formal comparison between the relation of brain activation to the Primacy versus the Recency mood models was conducted. We link the Primacy model to neural activity by a correlation between the model parameters and neural activation at the time preceding mood ratings. Specifically, we show that individual activation at the ACC and vmPFC is correlated to the weight of the expectation parameter of the Primacy model, but not the Recency model. These regions are implicated in mood regulation (*Bush et al., 2000*; *Etkin et al., 2015*; *Etkin et al., 2011*; *Hiser and Koenigs, 2018*; *Rudebeck et al., 2014*; *Stevens et al., 2011*; *Zald et al., 2002*) and in underlying decision making relative to previous outcomes (*Behrens et al., 2007*; *Scholl et al., 2017*; *Scholl et al., 2015*; *Wittmann et al., 2016*). Activity in these regions increased as the weight of the expectation parameter ($\beta_E$) of an individual was higher. Since the weight of this parameter determines the influence of previous outcomes on mood, this result suggests that these regions' activity plays a role in mediating the integration of previous outcomes to a subjective mood report. Therefore, the strength of this model-based fMRI analysis (*Cohen et al., 2017*; *O'Doherty et al., 2007*) is in allowing us to link neural signals to the computational relation between previous experiences and subjective mood reports.

Importantly, *Vinckier et al., 2018* also reported the mvPFC region as positively correlated in its activity level to changes in mood ratings, supporting the role that our model suggests for the mvPFC region in mediating mood ratings according to a primacy weighting of previous events. We did not find, however, the negative neural correlations to mood ratings that were also reported by that study.

Our experiments examine only a short space of time, no longer than 40 min. Human experiences are undoubtedly integrated over longer time periods, including temporally distant events in childhood. Whilst these are inherently difficult to model experimentally, it is noteworthy that early-life experiences, such as early adversity, are thought to exert long-term influences on mood (*Douglas et al., 2010*; *Lewis-Morrarty et al., 2015*; *Raby et al., 2015*). We also note that the time scales of our experiments are congruous to a number of real-life situations, both in research and clinical terms.

In research terms, self-reported mood in EMA is typically within the span of hours (*Kahneman et al., 2004*; *Larson et al., 1980*; *Taquet et al., 2020*). Given that the goal of EMA is often to uncover mood dynamics in relation to experiences in the environment, our results strongly indicate that explicit modeling of the relative timing of these two variables to each other may be crucial. Similarly, during fMRI and other scanning, researchers often ask participants to report on their mood during these sessions and use these to relate to neuroimaging results. Our results suggest that not just the value of events as such (whether, e.g., an aversive film was shown to participants), but also when it was shown may differentially impact such reports.

In terms of clinical events—such as patients' interactions with healthcare professionals for the purposes of psychotherapy or medication treatment—these typically last for about an hour. Importantly, the assessment of treatment progress relies on self-report (or clinician assessment of patients'

reports). Our results suggest that timing of such reports in relation to experiences during treatment could be an important source of variance.

Moreover, although our experiments test for different temporal structures of reward, they use a single type of task, a simple gambling decision task. It might be that the temporal structure of mood dependence is sensitive to the type of task or the context (i.e., social situations such as a conversation may include a different integration), which is an important matter for future studies. Another potential caveat relates to the online data collection using the Amazon Mechanical Turk (MTurk) platform, which can possibly include a different subset of participant characteristics (*Ophir et al., 2020*), but importantly, our results were robust to this difference and were well-replicated in our lab-based participants. In respect to the Primacy model characteristics, we aimed to minimize the divergence from the existing Recency model (we therefore changed the expectation term to consider the average of all previous outcomes but maintained the sum and the overall exponential discounting of that term). This computational modification between the models reflected our hypothesis that expectations in non-random temporal structures of rewards would be also influenced by the history of previous outcomes. This modification then resulted in a Primacy weighting. Nevertheless, the better performance of a Primacy weighting was consistent also when considering other formulations for weighing of previous events (and without taking into account the fewer parameters of the Primacy model).

Our results demonstrate that the Primacy model is superior to the Recency model, indicating that the full history of prior events influences mood. However, inclusion of a recency-weighted outcomes in the RPE term of the Primacy model prevents us from concluding simply that early events are more important than recent events in the ultimate outcome of self-reported mood. We therefore also note that when fitting the Primacy model the coefficients of the expectation term were significantly larger than the coefficients of the RPE term (which include recency-weighted outcomes), supporting the dominance of the expectation term primacy weighting (paired t-test with t = 2.6, p = 0.009, CI = [0.008,0.059]).

Additionally, there may be alternative, mathematically equivalent formulations of these models that would support different interpretations. Future work should compare the overall impacts of primacy and recency effects on mood with approaches robust to reparameterization, such as analysis of the causal effect of previous outcomes on mood using the potential-outcomes framework (*Imbens and Rubin, 2015*).

Overall, our conclusion that the effect of outcomes on mood through expectations has a primacy weighting in our tasks holds robustly when we consider a variety of different but similar models that either have primacy weighting (*Figure 3—figure supplement 2*) or recency weighting (*Table 1*). All the models with primacy weighting share that the expectation is based on an average over previous outcomes or potential outcome values. We stress that the expectation itself does not have to have primacy weighting for our conclusions to hold. The primacy model that we have chosen as our representative primacy model (due to having superior or statistically indistinguishable performance over the alternative primacy models) applies equal weights to all past outcomes to form the expectation, but we have also tried models where the weighting within the expectation had higher weights for more recent outcomes. In all these cases, the combination of current and past expectation still results in a primacy-weighted aggregate effect of previous outcomes on mood. The dependence of mood on an accumulation of previous expectations is therefore what causes the primacy weighting as the initial outcomes have a larger influence on mood versus a smaller influence of past expectation terms. In an intuitive sense, the primacy effect represents the greater weight first experiences have in a new environment or context, simply by virtue of coming first. The first event has nothing against which it can be compared, the second event has only itself and the first; the third event can be compared only against the first two, and so on, till eventually each additional event has a minimal impact in the face of all the events that have come before. The more trials we experience, the more information we gain, and the less meaning each event has on its own. This process has clear parallels to learning, but our models are agnostic to the exact mechanism by which expectations are accumulated. It is likely that there are equivalent formulations to our models in which expectation is a learned parameter controlled by a learning rate. The details of this mechanism are certainly of interest, but these will need to be elucidated by future studies.

More generally, our findings point to the importance of studying the temporal architecture of the interplay between experiences and mood. So far, computational and theoretic accounts of mood

have focused on event value (either in terms of expectations or outcomes or both) (*Eldar et al., 2016*; *Katsimerou et al., 2014*; *Russell et al., 1989*; *Rutledge et al., 2014*; *Vinckier et al., 2018*; *Watson and Tellegen, 1985*) as influences on subjective reports of well-being, while neglecting the importance of time. Our results suggest that in addition to these influential properties of the environment the dimension of time, that is, the temporal structure of previous events, also plays an important role, and that rather than being a matter of what happened most recently, the temporal representation of experience in mood seems to be dominated by a long-lasting effect of early events.

## Materials and methods

### 1. The Primacy versus Recency mood models

The formulation of both models consisted of two dynamic terms: the expectation term (E) and the RPE term (R), which is the difference of the outcome relative to the expected value.

Specifically, the Recency model of mood at trial t ($M_t$) was defined as

$$M_t = \epsilon_t + M_0 + \beta_C \sum_{j=1}^{t} \gamma^{t-j} C_j + \beta_E \sum_{j=1}^{t} \gamma^{t-j} E_j + \beta_R \sum_{j=1}^{t} \gamma^{t-j} R_j \tag{6}$$

where $\epsilon_t$ is a random noise variate with some unknown distribution (we may assume it to be normal with mean 0 and standard deviation $\sigma$), $M_0$ is the participant's baseline mood, $\gamma \in (0,1)$ is an exponential discounting factor, $C_j$ is the non-gamble certain amount at trial j (if not chosen then $C_j = 0$ and when chosen instead of a gamble then $E_j = R_j = 0$), $\beta_C$ is the participant's sensitivity to certain rewards during non-gambling trials, $\beta_E$ is the participant's sensitivity to expectation, and $\beta_R$ is the sensitivity to surprise during gambles.

In this model, the expectation term at trial t ($E_t$) was defined as the average between the two gamble values (see *Figure 1*, *Equation 2*) and the RPE term, R, was defined as

$$R_t = A_t - E_t \tag{7}$$

$A_t$ being the trial outcome.

In the Primacy model, the expectation term was replaced by the average of all previous outcomes (*Figure 1*, *Equation 3*) and R was defined similarly as shown in *Equation 7*. The overall Primacy model for mood at trial t was

$$M_t = \epsilon_t + M_0 + \beta_E \sum_{j=1}^{t} \gamma^{t-j} E_j + \beta_R \sum_{j=1}^{t} \gamma^{t-j} R_j \tag{8}$$

where $\beta_E$ and $\beta_R$ are the participant's sensitivity to previous outcomes and to how surprising these outcomes are relative to expectation, respectively. Note that this model performed better when we did not distinguish between gambling and non-gambling trials, which was another divergence from the standard Recency model.

### 2. Model fitting

All models were fit using a TensorFlow package (code can be downloaded using the link provided in Section 6). We chose group regularization constants by creating simulated datasets with realistic parameters and selecting the regularization parameters from a grid that had the best performance. The grid consisted of powers of 10 from 0.001 to 10,000.

For optimization, we used the following generic parametric model across subjects:

$$\hat{M}_s(t) = \mu_s + \sum_{v=1}^{p} \beta_{v,s} \sum_{j=1}^{t} \gamma_s^{t-j} X_{v,s}(j) \tag{9}$$

where s indexes the subject, $M_s(t)$ is subject's mood rating at trial t, $\mu_s$ is the subject-specific baseline mood, v is one of p time-varying task variables X (e.g., expectation or RPE values at each trial j),

and $\beta_{v,s}$ are subject-specific coefficients for each time-varying variable $X_{v,s}$ (note that we constrain $\beta_1, \ldots, \beta_3 \geq 0$).

To facilitate optimization, we further re-parameterized the discount factor $\gamma_s$ by defining

$$\gamma_s = \frac{1}{1 + \exp(-\xi_s)} \tag{10}$$

so that $\xi_s$ is an unbounded real number.

We found that the use of group-level regularization was necessary in order to stabilize the estimated coefficients. This took the form of imposing a variance penalty on $\xi$ and on each coefficient $\beta v$. The empirical variance is defined as

$$Var(\xi) = \frac{1}{n_s} \sum_{s=1}^{n_s} \left( \xi_s - \bar{\xi} \right)^2 \tag{11}$$

where $n_s$ is the number of subjects, and $\bar{\xi}$ is the group mean:

$$\bar{\xi} = \frac{1}{n_s} \sum_{s=1}^{n_s} \xi_s \tag{12}$$

We define Var($X_v$) for v = 1,…,p likewise.
The objective function is therefore

$$\text{minimize} \sum_{s=1}^{n_s} \sum_{t \in T} \left( \hat{M}_s(t) - M_s(t) \right)^2 + \lambda_\xi Var(\xi) + \lambda_\beta \sum_{v=1}^{p} Var(\beta_v) \tag{13}$$

where T is the set of trials where $M_s(t)$ was defined (optionally, one can also discard the first few trials in T to minimize window effects, we required $t \geq 11$), with $\lambda_\xi = 10$ and $\lambda_\beta = 100$ as the regularization parameters with the best performance in recovering the simulation ground truth for both models.

A leave-out sample validation approach was used in all model fitting, where a subsample of 40% randomly selected participants were modeled and then results were confirmed on the entire sample.

## 3. Testing the Primacy model across reward environments

### The random task

Participants played a gambling task where they experienced a series of different RPE values while rating their mood after every 2–3 trials. In this task, each trial consisted of a choice of whether to gamble between two monetary values or receive a certain amount. RPE values were randomly modified (ranging between −2.5 to +2.5) by assigning random values to the two gambles and a 50% probability for receiving one of these values as an outcome. The certain value was the average of the two gamble values. Specifically, each trial consisted of three phases: (1) gamble choice: 3 s during which the participants pressed left to get the certain value or right to gamble between two values (using a four-button response device); (2) expectation: only the chosen certain value or the two gamble options remained on the screen for 4 s; and (3) outcome: a feedback of the outcome value was presented for 1 s, followed by an inter-trial interval of 2–8 s. Participants completed 81 trials.

The mood rating consisted of two separate phases: (1) pre-rating mood phase, where the mood question 'How happy are you at this moment?' was presented for a random duration between 2.5 and 4 s, while the option to rate mood was still disabled; and (2) mood rating by moving a cursor along a scale labeled 'unhappy' on the left end and 'happy' on the right end. Each rating started from the center of the scale, and participants had a time window of 4 s to rate their mood. The cursor could move smoothly by holding down a single button press towards the left or the right directions. Each rating was followed by a 2–8 s jittered interval. Participants completed 34 mood ratings, and the overall task lasted 15 min.

### The structured task

In this version, participants experienced blocks of high or low RPE values, that is, patterns of positive or negative events, where RPE values were predefined and identical for all participants. RPE values

were set by a pre-made choice of the two gamble values and the outcome value, such that these values were random but RPE value, the difference between the average of the two values and the outcome, resulted in a predefined value (positive blocks of RPE = +5 during the first and the third blocks and a negative block in the middle of RPE = −5 during the second block). To maintain the unpredictability of outcomes within a block in the latter fixed version, 30% of the trials were incongruent to the block valence (i.e., small negative values of −1.5 during the first and third positive blocks, and a positive value of +1.5 during the second negative block).

The certain value was the average between the two gamble values. To avoid a predictable pattern of wins and losses, 30% of the trials were incongruent trials, where gamble and outcome values resulted in the opposite valence of RPE to the block (negative during the first and third positive blocks and positive during the middle negative block), but of a smaller value (RPE incongruent = 1.5). More specifically, the task consisted of three blocks, where each block had 27 trials and 11–12 mood ratings. Trials were identical to the random version in appearance. Participants again completed overall 34 mood ratings, and the overall task lasted 15 min.

## The structured-adaptive task

The structured-adaptive version was designed to maximally influence mood upwards or downwards by increasing or decreasing RPE values in real time. This task was identical to the structured task in the block design and number of trials but differed in RPE values being calculated in real time using a closed-loop control (PI algorithm used in control of nonlinear systems in engineering; *Levine, 2011*). Specifically, following each mood rating, RPE values were increased or decreased according to the difference of mood from the target mood, which was set to the highest mood value in the first and third positive blocks and to the lowest mood during the second block. This setup therefore generated personalized 'reward environments' as the task values were calculated online according to individual mood response and were not predetermined as in conventional paradigms.

More specifically, in each iteration of mood rating, the current mood, M(t), was compared to the block mood target value ($M_T$), which was set prior to the task in the aim to generate maximal mood transitions. Mood target value was defined as the maximal mood value on the mood scale in first and third blocks and the minimal mood value during the second block. To bring the mood value as close as possible to the target value $M_T$, the algorithm aimed at minimizing the error between the rated mood and the target mood value ($M_E$).

$$M_E(t+1) = \begin{cases} \dfrac{M_T - M(t)}{m_{max} - M_{min}}, & M_T = 100\% \text{ of mood scale} \\[2em] \dfrac{M(t) - M_T}{m_{max} - M_{min}}, & M_T = 0\% \text{ of mood scale} \end{cases} \tag{14}$$

The resulting $M_E$ value was between 0 and 1, then mapped to a change in the task RPE value, using a PI controller algorithm. This control algorithm uses a proportional and an integral error term derived from $M_E$. Importantly, the integral error term enables an RPE modification when mood remains in the same distance from the target mood value, and it was reset for each block.

Next, the RPE value was calculated such that the larger the mood error the stronger the modification of the RPE value, as follows:

$$RPE(t+1) = \begin{cases} RPE_{baseline} * M_E(t) + \sum_{1}^{t} M_E, & \text{Congruent trial} \\[2em] RPE_{baseline} * M_E(t) + \dfrac{\sum_{1}^{t} M_E}{3}, & \text{Incongruent trial} \end{cases} \tag{15}$$

where $RPE_{baseline}$ is a fixed value that was pre-calibrated to the value of 14 points (to have a moderate yet efficient influence of RPE change on mood). Congruent trials were 70% of trials, aligned with the control algorithm direction; the remaining 30% were incongruent, providing an RPE value with the opposite sign to the block context (set to be smaller in amplitude: on average, incongruent RPE values were −1.5 ± 0.8 SD). Then the two gamble values were calculated for the next 2–3 trials as follows:

$$L(t+1) = H(t+1) - RPE(t+1) \tag{16}$$

where H was the higher value, randomly assigned from a list ranging between $[-1.5, 14]$ with a step size of 0.2, and L is the lower gamble outcome. The allocation of these values between the upper or lower squares on the screen was randomly assigned.

The certain value (CR) that appeared on the left side was set to the average between the two values (unless this resulted in a certain value higher than two points, in which case it was half of the lower value $L(t + 1)$).

Last, the outcome value (A) was assigned according to the block $H(t + 1)$ in 70% of the first and third positive blocks trials, and $L(t + 1)$ during 70% of the second block trials (and vice versa in the 30% of trials that were incongruent).

This closed-loop circuit continued throughout the task, with each new mood rating used to update the reward values for the next series of 2–3 trials.

In all above task designs, participants were not informed of the probability of winning the gamble. We probed whether participants noticed that the win probability was rigged between blocks in the structured and structured-adaptive tasks with a follow-up questionnaire, which showed that most participants (90%, 65/72) were unaware of the manipulation (in a scale between 0 and 3, the average rating for whether the task was unfair was $0.36 \pm 0.69$ SD with 7/72 subjects indicating 'agree' or 'strongly agree').

## Participants

Participants completed either the random task (n = 60, mean age ± SD = 39.81 ± 13, 44% females), the structured task (n = 89, mean age ± SD = 37.55 ± 10.46, 44% females), or the structured-adaptive task (n = 80, mean age ± SD = 37.76 ± 11.23, 42% females). See *Table 2* for participants characteristics.

These participants were recruited from Amazon Mechanical Turk (MTurk) system and completed the tasks online. Analyses of this structured-adaptive dataset were publicly preregistered on an open science online repository to confirm our modeling results (https://osf.io/g3u6n/). The MTurk Worker ID was used to distribute a compensation of $8 for completing the task and a separate task bonus between $1 and $6 according to the points gained during the task. Participants were instructed before the task that they would receive a payment that is proportional to the points that they gain during the task. These study populations were ordinary, non-selected adults of 18 years of age or older. Participants were not screened for eligibility, all individuals living in the US and who wanted to participate were able to do so. Participants were restricted to doing the task just once. Three participants were excluded from analyses due to an error in the task script where mood ratings were inconsistently spread along the three blocks. All participants received similar scripted instructions and provided informed consent to a protocol approved by the NIH Institutional Review Board.

## Statistical testing of the influence of reward environments on mood

We applied a linear mixed effects model to estimate the task influence on mood using the nlme package in RStudio (2020). This model enabled the estimation of the across-participants significance of mood change while controlling for the within-participant variability in mood change slopes and intercepts, defined as random effects. Specifically, the independent variable was the response variable of interest mood (M), and the dependent variables were time (t, which is the trial index) and time squared (t2), with the two different time variables considered as random effects, as follows:

$$M \sim t + t2 + (t + t2 | \text{subject}) \tag{17}$$

The effect was considered significant with $p < 0.05$. All t-tests conducted were two-sided.

## 4. Testing the Primacy model across participant characteristics and neural signals

An additional dataset of the structured-adaptive task was collected in an fMRI scanner, providing us with different experimental conditions, a different age group of adolescent participants, data of

**Table 2.** Participants' demographics for all datasets.

| Random online MTurk sample | Age | |
|---|---|---|
| (n = 67) | Average | 39.81 |
| | SD | 13 |
| | **Sex** | |
| | Male | 37 |
| | Female | 32 |
| **Structured online MTurk sample** | **Age** | |
| (n = 89) | Average | 37.55 |
| | SD | 10.46 |
| | **Sex** | |
| | Male | 48 |
| | Female | 41 |
| **Structured-adaptive online MTurk sample** | **Age** | |
| (n = 80) | Average | 37.76 |
| | SD | 11.23 |
| | **Sex** | |
| | Male | 46 |
| | Female | 34 |
| **Structured-adaptive lab-based sample** | **Age** | |
| (n = 72) | Average | 15.49 |
| | SD | 1.48 |
| | **Sex** | |
| | Male | 17 |
| | Female | 55 |
| | **MFQ score** | |
| | Average | 5.81 |
| | SD | 5.98 |
| | **Diagnosis** | |
| | Healthy volunteer | 29 |
| | MDD | 43 |

participants diagnosed with depression, and a recording of neural signals during the task (n = 72, mean age ± SD = 15.49 ± 1.48, 76% females, mean depression score MFQ ± SD = 5.81 ± 5.98, n = 43 participants met diagnostic criteria for depression according to DSM-5, of whom at the time of the experiment n = 18 had an ongoing depressive episode and n = 35 were medicated). See *Table 2* for participants characteristics. These participants completed the task in an fMRI scanner and were compensated for doing the task and for scanning, as well as receiving a separate bonus proportional to the points earned during the task (a value between $5 and $35). This task version lasted 24 min instead of the duration of 15 min of the online versions to allow for an optimal analysis of brain data. Participants were screened for eligibility, and inclusion criteria were the capability to be scanned in the MRI scanner and not satisfying diagnosis criteria for disorders other than depression according to DSM-5. Overall, five participants were excluded from analyses due to incomplete data files, and three additional participants were excluded due to repeatedly rating a single fixed mood value for an entire block of the task. These participants received the same scripted instructions and provided informed consent to a protocol approved by the NIH Institutional Review Board.

## 5. Analyzing the neural correlates of the Primacy model

### fMRI data acquisition

Participants in the adolescent sample performed the structured-adaptive task while scanning in a General Electric (Waukesha, WI) Signa 3-Tesla MR-750s magnet, being randomly assigned to one of two similar scanners. Task stimuli were displayed via back-projection from a head-coil-mounted mirror. Foam padding was used to constrain head movement. Behavioral choice responses were recorded using a hand-held Fiber Optic Response Pad (FORP). 47 oblique axial slices (3.0 mm thickness) per volume were obtained using a T2-weighted echo-planar sequence (echo time, 30 ms; flip angle, 75°; 64 × 64 matrix; field of view, 240 mm; in-plane resolution, 2.5 mm × 2.5 mm; repetition time was 2000 ms). To improve the localization of activations, a high-resolution structural image was also collected from each participant during the same scanning session using a T1-weighted standardized magnetization prepared spoiled gradient recalled echo sequence with the following parameters: 176 1 mm axial slices; repetition time, 8100 ms; echo time, 32 ms; flip angle, 7°; 256 × 256 matrix; field of view, 256 mm; in-plane resolution, 0.86 mm × 0.86 mm; NEX, 1; bandwidth, 25 kHz. During this structural scanning session, all participants watched a short neutral-mood documentary movie about bird migration.

### Data preprocessing

Analysis of fMRI data was performed using Analysis of Functional and Neural Images (AFNI; *Cox, 1996*) software (version 19.3.14). Standard preprocessing of EPI data included slice-time correction, motion correction, spatial smoothing with a 6 mm full-width half-maximum Gaussian smoothing kernel, normalization into Talairach space and a 3D nonlinear registration. Each participant's data were transformed to a percent signal change using the voxel-wise time-series mean blood oxygen-level-dependent (BOLD) activity. Time series were analyzed using multiple regression (*Neter et al., 1990*), where the entire trial was modeled using a gamma-variate basis function. The model included the following task phases: gamble choice: an interval that lasted up to 3 s, from the presentation of the three monetary values to the choice button press, left for the certain amount or right to gamble. Expectation: a 4 s interval from making the choice of whether to gamble to receiving the gamble outcome. Outcome: a 1 s interval during which the received outcome is shown. The pre-rating interval: a variable interval between 2.5 and 4 s when the mood question is presented but the option to rate mood is still disabled. Mood rating phase: a 4 s interval during which participants rate their mood. The model also included six nuisance variables modeling the effects of residual translational (motion in the x, y, and z planes), rotational motion (roll, pitch, and yaw), and a regressor for baseline plus slow drift effect, modeled with polynomials (baseline being defined as the non-modeled phases of the task). Echo-planar images (EPIs) were visually inspected to confirm image quality and minimal movement. The code for generating the full processing stream for each participant was created using the afni_proc.py command. This script creates also a quantitative and qualitative quality control (QC) outputs, which were used to verify the processing in the present study. We then ran a whole-brain, group-level ANOVA (3dMVM [*Chen et al., 2014*] in AFNI) with the weights of the Primacy or the Recency model as between-participant covariates of each of these neural activations (each participant's neural activity was represented by a single whole-brain image of activation across all trials).

### Statistical significance

This was determined at the group level using 3dClustSim (the latest acceptable version in AFNI with an ACF model), which generated a corrected to $p<0.05$ voxel-wise significance threshold of $p<0.005$ and a minimal cluster size of 100 voxels. We analyzed relation to model parameters with neural activity during three different phases of the task: activation during the pre-mood rating period, mood rating encoding (with mood values as a parametric regressor of the mood pre-rating period), and task-based RPE encoding (RPE values as a parametric regressor of the outcome period). Since these are three separate tests, we added a Bonferroni correction to the multiple comparison correction, which resulted in a final p-value threshold of $0.005/3 = 0.0017$.

## 6. Code and data availability

To enable the reproducibility of this study we made all scripts and datasets available online at: https://osf.io/vw7sz. This online repository includes the scripts for modeling the mood data, the source-data of *Figure 2* (tasks and mood rating data of all participants), the afni_proc and 3dMVM neural analysis scripts and the whole-brain neural images presented in *Figure 4*. All the unprocessed neuroimaging data can be found online at: https://openneuro.org/datasets/ds003709.

## Acknowledgements

We thank Elisabeth A Murray and Nathaniel D Daw for helpful comments and questions. This research was supported in part by the Intramural Research Program of the National Institute of Mental Health National Institutes of Health (NIH) (Grant No. ZIA-MH002957-01 to AS). RBR is supported by the National Institute of Mental Health (1R01MH124110), a Medical Research Council Career Development Award (MR/N02401X/1), and a NARSAD Young Investigator Award from the Brain & Behavior Research Foundation, P&S Fund. This work used the computational resources of the NIH HPC (high-performance computing) Biowulf cluster (http://hpc.nih.gov). The funder had no role in the design and conduct of the study; collection, management, analysis, and interpretation of the data; preparation, review, and approval of the manuscript; or decision to submit the manuscript for publication. The views expressed in this article do not necessarily represent the views of the National Institutes of Health, the Department of Health and Human Services, or the United States Government.

## Additional information

### Funding

| Funder | Grant reference number | Author |
|---|---|---|
| National Institute of Mental Health | Intramural Research Program no. ZIAMH002957-01 | Hanna Keren Katharine Chang Aria Vitale Dylan Nielson Argyris Stringaris |
| National Institute of Mental Health | Intramural Research Program | Charles Zheng David C Jangraw Francisco Pereira |
| National Institute of Mental Health | | Robb B Rutledge |
| Medical Research Council | Career Development Award MR/N02401X/1 | Robb B Rutledge |
| Brain and Behavior Research Foundation | NARSAD Young Investigator Award | Robb B Rutledge |

The funders had no role in study design, data collection and interpretation, or the decision to submit the work for publication.

### Author contributions

Hanna Keren, Conceptualization, Software, Formal analysis, Investigation, Visualization, Methodology, Writing - original draft, Writing - review and editing; Charles Zheng, Conceptualization, Software, Formal analysis, Methodology, Writing - review and editing; David C Jangraw, Software, Project administration; Katharine Chang, Aria Vitale, Project administration; Robb B Rutledge, Conceptualization, Writing - review and editing; Francisco Pereira, Supervision, Writing - review and editing; Dylan M Nielson, Conceptualization, Supervision, Writing - review and editing; Argyris Stringaris, Conceptualization, Supervision, Methodology, Writing - original draft, Writing - review and editing

## Author ORCIDs

Hanna Keren ![ORCID] https://orcid.org/0000-0003-4122-656X
Robb B Rutledge ![ORCID] http://orcid.org/0000-0001-7337-5039

## Ethics

Clinical trial registration NCT03388606.
Human subjects: All participants signed informed consent to a protocol approved by the NIH Institutional Review Board. The protocol is registered under the clinical trial no. NCT03388606.

## Decision letter and Author response

Decision letter https://doi.org/10.7554/eLife.62051.sa1
Author response https://doi.org/10.7554/eLife.62051.sa2

## Additional files

### Supplementary files

• Supplementary file 1. Model parameters recovery analysis: results of fitting the Primacy and Recency models on simulated datasets as well as a statistical comparison between the two models in both the training errors and the streaming prediction errors. Note that only streaming prediction errors were used for model selection, and we show the training errors for illustrative purposes.

• Supplementary file 2. The formulation of alternative variants of the Recency model.

• Transparent reporting form

### Data availability

To enable the reproducibility of this study we made scripts and datasets available online at: https://osf.io/vw7sz/. This repository includes: Mood modeling code; Source-data of Figure 2 (tasks trial-wise values and mood ratings values of all participants); Neural analyses code; Files of the whole-brain neural images presented in Figure 4. All the unprocessed neuroimaging data can be found online at: https://openneuro.org/datasets/ds003709.

The following dataset was generated:

| Author(s) | Year | Dataset title | Dataset URL | Database and Identifier |
|---|---|---|---|---|
| Keren H | 2020 | Primacy Weight on Mood - Analyses code and datasets | https://osf.io/vw7sz | Open Science Framework, vw7sz |

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
