## [Decision Letter]

**Acceptance summary:**

This is a very interesting study whose goal is to determine what drives subjective mood over time during a reward-based decision making task. The authors report data from a series of online behavioural studies and one performed during neuroimaging. Participants played a well-established gambling task during which they had to select between a sure outcome and a 50:50 gamble, reporting momentary mood assessments throughout the game. The authors compared the performance of a number of computational models of how the mood ratings were generated.

The authors identify as their "baseline" model that proposed by Rutledge and colleagues, in which an important determinant of mood seems to be the reward prediction error - this is called the Recency model. They contrast it with a Primacy model, in which earlier events (in this case, aggregated and weighted experienced outcomes) play a more important role. They validate the model across different behavioural conditions, involving healthy participants and depressed patients. The conclusion is that the data are more consistent with their Primacy model, in other words a stronger weight of earlier events on reported mood. In the fMRI experiment they found that the weights of the Primacy model correlated with prefrontal activation across participants, while this was not the case for the Recency model.

The paper is clearly written and easy to understand. The question of how humans combine experienced events into reported mood is topical and the conclusions are striking, given the dominance of recency-based models in the literature (e.g. Kahneman's peak-end heuristic). The paper takes an interesting approach and presents an impressive amount of data, and the reviewers and editor felt that it makes a substantial contribution to the literature.

**Decision letter after peer review:**

Thank you for submitting your article "The temporal representation of experience in subjective mood" for consideration by eLife. Your article has been reviewed by 3 peer reviewers, and the evaluation has been overseen by a Reviewing Editor and Timothy Behrens as the Senior Editor. The reviewers have opted to remain anonymous.

The reviewers have discussed the reviews with one another and the Reviewing Editor has drafted this decision to help you prepare a revised submission.

As the editors have judged that your manuscript is of interest, but as described below that substantial additional analyses are required before it is published, we would like to draw your attention to changes in our revision policy that we have made in response to COVID-19 (https://elifesciences.org/articles/57162). First, because many researchers have temporarily lost access to the labs, we will give authors as much time as they need to submit revised manuscripts. We are also offering, if you choose, to post the manuscript to bioRxiv (if it is not already there) along with this decision letter and a formal designation that the manuscript is "in revision at eLife". Please let us know if you would like to pursue this option. (If your work is more suitable for medRxiv, you will need to post the preprint yourself, as the mechanisms for us to do so are still in development.)

Summary:

This is a very interesting study whose goal is to determine what drives subjective mood over time during a reward-based decision making task. The authors report data from a series of online studies and one performed with fMRI. Participants played a well-established gambling task during which they had to select between a sure outcome and a 50:50 gamble, reporting momentary mood assessments throughout the game. The authors compared the performance of a number of models of how the mood ratings were generated.

The authors identify as their "baseline" model that proposed by Rutledge and colleagues, in which an important determinant of mood seems to be the reward prediction error - the authors call this Recency model. They contrast it with a Primacy model, where earlier events (in this case, average experienced outcomes) play a more important role. They validate the model across different behavioural conditions, involving healthy subjects, teenagers and depressive patients. The conclusion is that the data are more consistent with their Primacy model, in other words a higher weight of earlier events on reported mood. In the fMRI experiment they found that the weights of the Primacy model correlated with prefrontal activation across subjects, while this was not the case for the Recency model.

The paper is clearly written and easy to understand. The question of how humans combine experienced events into reported mood is topical and the conclusions are striking, given the dominance of recency-based models in the literature (e.g. Kahneman's peak-end heuristic). The paper takes an interesting approach and presents an impressive amount of data.

However, at some points the arguments seemed a considerable stretch, in part because important experimental and methodological detail is missing, and in part because the analyses do not currently consider a number of potential confounds in both the models and the task design. It is not clear whether these concerns can be addressed or not, but we would like to give you the opportunity to do so. Ultimately, these concerns come down to whether we can be certain that the results reflect a true primacy effect, as opposed to some other process that simply appears at face value to be a primacy effect.

To this end, some important checks need to be made concerning both the computational and the fMRI analyses, as detailed below. These do require substantial extra modelling work, and it is quite possible that the conclusions will not survive these control analyses.

Essential revisions:

1. In relation to model comparison, the authors need to show us whether or not their model selection criteria allow them to correctly recover the true generative model in simulated datasets. Are we sure that the model selection criteria are unbiased toward the two models?

2. Related to point (1), can the authors provide a qualitative signature of mood data that falsifies the Recency model at the group level (see Palminteri, Wyart and Koechlin. 2017). They do so in Figure S2 for one participant, but it would be important to show the same (or similar) result at the group level (this should be easier in the structured or in the structured-adaptive conditions).

3. It is not clear where the weights on the primacy graph (Figure 1B) come from. The recency weights make sense - there is a discount factor in the model that is less than 1, so there is a an exponential discount of more distant past events. However, for the primacy model the expectation is apparently calculated as the arithmetic mean of previous outcomes (which suggests a flat weight across previous trials) and the discount factor remains. So how can this generate the decreasing pattern of weights? It would be really useful if the authors could spell this out as it is currently quite confusing.

4. The models differ in terms of whether they learn about the expected value of the gamble outcomes, or whether they assume a 50:50 gamble (the Recency model assumes this, the Primacy model generates an average of all experienced outcomes). This leaves open the possibility that the benefit of the Primacy model is simply that people do in fact largely use experienced outcomes to generate their expectations, rather than believing the outcome probabilities displayed in the experiment. Can the authors exclude this possibility?

5. Related to the above point, the structured and adaptive environments seem to have something to learn about (blocks with positive vs. negative RPEs), so it is perhaps not surprising that humans show evidence of learning here, and a model with some learning outperforms one with none.

The description of these environments is insufficient at present - can the authors explain how RPEs were manipulated? Was it by changing the probability of win/loss outcomes, and if so how? Or was it by changing the magnitudes of the options? For the adaptive design was the change deterministic? And was the outcome (and thus the RPE) therefore always positive if mood was low; or was this probabilistic, and if so with what probability? Finally, did the Recency model still estimate its expectations here as 50:50, even when this was not the case? If so, this requires justification.

6a. In addition to changing the expectation term of the Recency model, the Primacy model also drops the term for the certain outcomes (because this improves model performance). Can this account for the relative advantage of the Primacy over the Recency model? In other words, if the certain outcome term is dropped from the Recency model as well, does the Primacy model still win? If the authors want to establish conclusively that Primacy is a better model than Recency, then surely more models ought to be compared, at the very least using a 2x2 design with primacy/recency of expectations/outcomes.

6b. On a related point, the standard Recency model was originally designed such that the certain option C was NOT the average of the two gambles, so C was required in the model (at least in the 2014 PNAS paper). Here, C is the average of the gambles, so presumably it would be identical to E in the Recency model, and therefore be extraneous in the Recency model as well as the Primacy model. Did the authors perform model comparison to see if C could be eliminated from the Recency model? If so, this is not another difference between the models after all.

7. The structured and structured-adaptive tasks seem to have some potential problems when it comes to assessing their impact on mood ratings:

i. the valence of the blocks was not randomised, meaning that the results could be confounded. E.g. what if negative RPE effects are longer-lasting than positive RPE effects? This seems plausible given the downward trend in mood in the random environment despite average RPE of zero. Could this also explain the pattern of mood in the other two tasks, rather than primacy?

ii. scaling: if there is a marginally decreasing relationship between cumulative RPE and mood (such that greater and greater RPEs are required to lift/decrease mood by the same amount), then this will resemble a primacy effect? This is unlikely to be an issue in the random task but could be a problem in the structured and certainly in the structured-adaptive tasks.

iii. individual differences in responsiveness to RPE: in the structured-adaptive task, some subjects' mood ratings may be very sensitive to RPE, and others very insensitive. One might expect that given the control algorithm has a target mood, the former group would reach this target fairly soon and then have trials without any RPEs, while the latter group would not reach the target despite ever increasing RPEs. In both cases the Primacy model would presumably win, due to sensitivity to outcomes in the first half or insensitivity to bigger outcomes in the second half respectively? Can the authors exclude these possibilities?

8. In relation to the fMRI analyses, the results in the main text seem to result from a second-level ANCOVA, where the individual weights of the Primacy model are shown to correlate with activation in the prefrontal cortex. Similar analyses using the weights of Recency model do not produce significant results at the chosen threshold. This analysis is problematic for two reasons. First, absence of evidence does not imply evidence of absence - was a formal comparison of the regression coefficients conducted? Second, to really validate the model the authors should show that the trial-by-trial correlates of expectations and prediction errors are more consistent with the Primacy than the Recency model, using a parametric analysis at the participant level.

9. Similar to point 6, it is hard to conclude much about the models from the fact that the Primacy model E beta (but not the Recency model E beta) correlates with BOLD responses in a prefrontal cluster, when the Recency model E term is based on previous expectations, not previous outcomes. Likewise with the direct comparison of the models' voxel-wise correlation images.

[Editors' note: further revisions were suggested prior to acceptance, as described below.]

Thank you for submitting your article "The temporal representation of experience in subjective mood" for consideration by eLife. Your article has been reviewed by the same 3 peer reviewers, and the evaluation has been overseen by a Reviewing Editor and Timothy Behrens as the Senior Editor. The reviewers have opted to remain anonymous.

The reviewers have discussed their reviews with one another, and the Reviewing Editor has drafted the below summary to help you prepare a revised submission. The reviewers felt that you had been very receptive to their comments, making extensive changes and performing the requested additional analyses. Consequently, the manuscript is nearly ready for publication, subject to a few final clarifications that are listed below.

Essential Revisions:

1. The new Figure S1 is helpful, but caused some confusion as the description in the legend could be clearer. The reviewers assume that the blue bars are the weights attached to all outcomes (not expectations) from trial 1-8, all contributing to the expectation on trial 8; the orange bars are the weights attached to all outcomes from trial 1-7, all contributing to the expectation on trial 7; and so on. Assuming this understanding is correct, please amend the text in the legend to clarify this better, and also provide a colour key for the figure to make it clear that the bars refer to outcomes from specific trials.

2. The Primacy vs Recency model comparison is of critical importance, so it would help the paper to be as clear as possible about it. The key comparison here is between the variant of the Recency model that is identical to the Primacy model in terms of having both a learned outcome as the expectation and without the certainty term. Your model comparison seems to test these changes separately rather than together (apologies if we have misunderstood this).

Can you explain what it is, specifically, about the Primacy model that causes it to perform better than the best Recency one (e.g. is it the nature of the way the expectation is learned, or something else)? As we understand it, the central claim of your paper is that people learn the expectation term in a way that effectively puts more weight on the initial outcomes encountered, so it would be really useful to understand what it is about the learning process that causes this effect, and whether it is this that improves the fit of the model.

3. Relating to the above point, the key comparisons between Primacy and Recency models in the main manuscript (e.g. the comparative fMRI analyses) should be between these similar models, not the Primacy and original Recency model.

---

## [Author Response]

Essential revisions:1. In relation to model comparison, the authors need to show us whether or not their model selection criteria allow them to correctly recover the true generative model in simulated datasets. Are we sure that the model selection criteria are unbiased toward the two models?

Thank you for asking for this validation. It has helped us to finetune our model selection criteria and reach a single optimal predictive criterion (as described below) that indeed allows us to recover the true generative model in simulated data. Using this selection criterion, we replicated our finding that the Primacy model was a better temporal description of mood ratings, and in addition, we use this criterion to test a set of new models that we developed to address the questions raised by the reviewers.

Method:

We performed a model recovery assessment to validate our model selection criteria comparing the Primacy and Recency models (Wilson and Collins, 2019). In order to simulate the mood ratings, we first fit the Primacy and Recency models to the Structured task data. We then used the variance of the residuals as the variance for Gaussian distributions around the predicted mood values produced by each model. Next, we generated simulated data for each participant for each model based on these distributions. Finally, we fit the models to both sets of simulated data and evaluated the ability of each of our model selection criteria to correctly identify the model that generated the data. Significant differences for each criterion were evaluated with the Wilcoxon test, as was done for our main analysis.

Results:

According to both the training error and the streaming prediction criteria, the Recency model performs better on data simulated with the Recency model and the Primacy model performs better on data simulated with the Primacy model (see Table S1). In the process of conducting the model recovery assessment, we found that the streaming prediction error criterion was unstable in the first few trials due to fewer available data points, so we modified this criterion to discard the first ten mood ratings. We then opted to use the streaming prediction errors for model comparison, as it is a more valid criterion due to the training error favoring overfitting (as a model with more parameters will fit the training data better). Other performance tests, such as the AIC and BIC, do place a penalty on each additional parameter added to the model. But, on the other hand, the AIC still tends to favor overfitting while the BIC tends to be conservative. Furthermore, both AIC and BIC require the assumption of a parametric distribution for the noise term, while metrics based on prediction error do not require such an assumption. Therefore, we could apply streaming prediction directly to our models while application of AIC or BIC would have required us to make additional assumptions about the distributional form of the noise terms. The streaming prediction error we use is a held-out prediction error, reflecting sequential prediction of each mood rating (Hastie, Tibshirani et al., 2009). We therefore used the streaming prediction criterion to replicate our previous result as well as to test the newly developed models.

It is not a cross-validation, which is unfeasible because of the temporal dependency in our data.

Changes made:

On page 9 of the results:

“The Primacy vs Recency models performance criteria

We then performed a model recovery assessment to validate the model selection criteria by which we compare the performance of the Primacy and Recency models. We first generated simulated datasets using each of the models and then fit the models and tested whether we could correctly identify the model that generated the data. According to both the training error and the streaming prediction criteria, it was possible to recover the true model from the simulated data (the Recency model performed better on data simulated with the Recency model, and vice versa, see Table-S1). We then preferred to use the streaming prediction error across all comparisons, as it is a more valid criterion due to the training error favoring overfitting (Hastie, Tibshirani et al., 2009).”

On Page 5, of the Introduction:

“We then examine the generalizability of the performance of the two models across simulated data, in a model recovery analysis that protects from model selection biases (Hastie, Tibshirani et al., 2009).”

2. Related to point (1), can the authors provide a qualitative signature of mood data that falsifies the Recency model at the group level (see Palminteri, Wyart and Koechlin, 2017). They do so in Figure S2 for one participant, but it would be important to show the same (or similar) result at the group level (this should be easier in the structured or in the structured-adaptive conditions).

We have extended our Figure S2 (now Figure S4 on page 48) to include a group level representation of mood ratings and the respective trial-level Recency and Primacy model parameters:

3. It is not clear where the weights on the primacy graph (Figure 1B) come from. The recency weights make sense - there is a discount factor in the model that is less than 1, so there is an exponential discount of more distant past events. However, for the primacy model the expectation is apparently calculated as the arithmetic mean of previous outcomes (which suggests a flat weight across previous trials) and the discount factor remains. So how can this generate the decreasing pattern of weights? It would be really useful if the authors could spell this out as it is currently quite confusing.

This point is a very important one to make clear to the reader and therefore we thank the reviewers for asking us to clarify.

We now further explain in the paper (on page 6) that the stronger weight of earlier outcomes in the Primacy model emerges from two separate aspects of the model: First, that one’s expectation for the next outcome is based on the average of all previously received outcomes and, moreover, that mood is determined by the sum of all such past expectations. We also illustrate these two aspects of the model and how they are integrated to a Primacy weighting with the following graphical illustration (on Page 40):

4. The models differ in terms of whether they learn about the expected value of the gamble outcomes, or whether they assume a 50:50 gamble (the Recency model assumes this, the Primacy model generates an average of all experienced outcomes). This leaves open the possibility that the benefit of the Primacy model is simply that people do in fact largely use experienced outcomes to generate their expectations, rather than believing the outcome probabilities displayed in the experiment. Can the authors exclude this possibility?

To test the possibility that the better performance of the Primacy model is due to individuals using experienced outcomes to generate their expectations, rather than the Primacy weighting, we compared a new model against the Primacy model: a Recency model where the expectation term is based on the previous outcome rather than current trial’s possible outcome values (the “Recency with outcome as expectation model”).

This test showed that the Primacy model performed better than this “Recency with outcome as expectation model”, in the random, Structured, and Structured-adaptive tasks, according to the streaming prediction performance criterion. This is also the result for the additional sample of adolescents, that includes clinically depressed subjects as well as different experimental conditions (lab-based rather than online collection). See Table-1 for a summary of these results.

We further addressed this possibility along with the possible learning component of the Primacy model in the Structured and Structured-adaptive tasks by creating a Recency model that considers the actual individual winning probability, instead of a fixed win probability of 50% The trial-level individual winning probability is the percentage of previous trials with a win outcome. This is calculated for each trial by dividing the number of preceding trials that resulted in winning the higher outcome by the overall number of previous trials. This model also introduces a similar component of learning from previous trials to the Recency model, allowing us to test whether this “learning” accounts for the superior performance of the Primacy model, as suggested in point (5).

This “Dynamic win probability recency model” was implemented by considering win probability (Equation 4):

where the sum in the numerator counts the previous trials on which the higher value H was the outcome (I is a binary vector of this condition with 1 for the outcome H and 0 for the lower outcome L) and the sum in the denominator counts the previous trials on which the participant chose to gamble (G is a binary vector of this condition with 1 for the choice to gamble and 0 for the choice of the certain value). The additional bias of 5 in the numerator and 10 in the denominator implement Bayesian shrinkage corresponding to 10 prior observations with an average success of 0.5.

The expectation term uses the dynamically updated win probability to compute expected gambling outcome (Equation 5):

This test showed that the Primacy model performed better than this dynamic Recency model, for all three Random, Structured and Structured-adaptive reward tasks, according to the validated streaming prediction criterion (see Table-1 for stats).

Changes made:

We summarize these new results in Table-1 which presents all model fitting values and stats for each of the three reward environments:

We describe these new results in the added subsection ““Primacy vs other variants of the Recency model” of the Results on page 14:

In addition, these results are presented in the new supplemental Table-S2 on page 46 which describes the formulations of the new alternative Recency models:

And in the new Figure 3B on page 17:

We also added another section to the discussion, where we address additional aspects of the Primacy model on page 25:

“Our results demonstrate that the Primacy model is superior to the Recency model, indicating that the full history of prior events influences mood. However, inclusion of a recency-weighted outcomes in the RPE term of the Primacy model prevents us from concluding simply that early events are more important than recent events in the ultimate outcome of self-reported mood. We therefore also note, that when fitting the Primacy model the coefficients of the expectation term were significantly larger than the coefficients of the RPE term (which include recency weighted outcomes), supporting the dominance of the expectation term primacy weighting (paired t-test with t = 2.6, p = 0.009, CI = [0.008,0.059]). Additionally, there may be alternative, mathematically equivalent formulations of these models that would support different interpretations. Future work should compare the overall impacts of primacy and recency effects on mood with approaches robust to reparameterization, such as analysis of the causal effect of previous outcomes on mood using the potential-outcomes framework (Imbens and Rubin, 2015).”

5. Related to the above point, the structured and adaptive environments seem to have something to learn about (blocks with positive vs. negative RPEs), so it is perhaps not surprising that humans show evidence of learning here, and a model with some learning outperforms one with none.The description of these environments is insufficient at present - can the authors explain how RPEs were manipulated? Was it by changing the probability of win/loss outcomes, and if so how? Or was it by changing the magnitudes of the options? For the adaptive design was the change deterministic? And was the outcome (and thus the RPE) therefore always positive if mood was low; or was this probabilistic, and if so with what probability? Finally, did the Recency model still estimate its expectations here as 50:50, even when this was not the case? If so, this requires justification.

We have added elaborate explanations of the different reward environments, addressing both the manipulation of RPEs in general as well as specifically in the adaptive task.

We have extended and modified the task descriptions as follows (starting on page 30 of the Methods section):

“The Random task:

Participants played a gambling task where they experienced a series of different RPE values while rating their mood after every 2-3 trials. In this task, each trial consisted of a choice of whether to gamble between two monetary values or receive a certain amount. RPE values were randomly modified [ranging between -2.5 to +2.5] by assigning random values to the two gambles and a 50% probability for receiving one of these values as an outcome. The certain value was the average of the two gamble values.”

“The Structured task:

In this version participants experienced blocks of high or low RPE values, i.e., patterns of positive or negative events, where RPE values were pre-defined and identical for all participants. RPE values were set by a pre-made choice of the two gamble values and the outcome value, such that these values were random but RPE value, the difference between the average of the two values and the outcome, resulted in a pre-defined value (positive blocks of RPE=+5 during the first and the third blocks and a negative block in the middle of RPE = –5 during the second block). To maintain the unpredictability of outcomes within a block in the latter Fixed version, 30% of the trials were incongruent to the block valence (i.e., small negative values of -1.5 during the first and third positive blocks, and a positive value of +1.5 during the second negative block).

The certain value was the average between the two gamble values. To avoid a predictable pattern of wins and losses, 30% of the trials were *incongruent trials*, where gamble and outcome values resulted in the opposite valence of RPE to the block (negative during the first and third positive blocks and positive during the middle negative block), but of a smaller value (RPE incongruent = 1.5). More specifically, the task consisted of three blocks, where each block had 27 trials and 11-12 mood ratings. Trials were identical to the Random version in appearance. Participants again completed overall 34 mood ratings in 15 min of task time.”

"The Structured-Adaptive task:

More specifically, in each iteration of mood rating, the current mood, M(t), was compared to the block mood target value (M_T_), which was set prior to the task in the aim to generate maximal mood transitions. Mood target value was defined as the maximal mood value on the mood scale in first and third blocks, and the minimal mood value during the second block. To bring the mood value as close as possible to the target value M_T_, the algorithm aimed at minimizing the error between the rated mood and the target mood value (M_E_).

[…]

This closed-loop circuit continued throughout the task, with each new mood rating used to update the reward values for the next series of 2-3 trials.”

The structured-adaptive task design was probabilistic as there was a 70% chance for a congruent trial.

As for the second part of the question (“Finally, did the Recency model still estimate its expectations here as 50:50, even when this was not the case”):

The Recency model indeed considered 50-50 win probability, which was not the case in this task design. We therefore validated our result on the random task design, where the win probability was 50-50. We also validated the Primacy model against a Recency model with a dynamic win probability (see point 4 above).

6a. In addition to changing the expectation term of the Recency model, the Primacy model also drops the term for the certain outcomes (because this improves model performance). Can this account for the relative advantage of the Primacy over the Recency model? In other words, if the certain outcome term is dropped from the Recency model as well, does the Primacy model still win? If the authors want to establish conclusively that Primacy is a better model than Recency, then surely more models ought to be compared, at the very least using a 2x2 design with primacy/recency of expectations/outcomes.6b. On a related point, the standard Recency model was originally designed such that the certain option C was NOT the average of the two gambles, so C was required in the model (at least in the 2014 PNAS paper). Here, C is the average of the gambles, so presumably it would be identical to E in the Recency model, and therefore be extraneous in the Recency model as well as the Primacy model. Did the authors perform model comparison to see if C could be eliminated from the Recency model? If so, this is not another difference between the models after all.

To answer this question, we have conducted the suggested analysis and dropped the Certain outcome term from the recency model, to form the model “Recency without a Certain term”.

We found that the Primacy model performed better than this alternative model for all three Random, Structured and Structured-adaptive reward tasks according to the validated streaming prediction criterion (see Table-1 for stats). This model is described in the new subsection of the Results on page 14.

We appreciate the contribution of the reviewers in pointing out these alternatives and hope that our results presented in response to this item and the previous items demonstrates that the Primacy effect is unlikely to be a result of these alternative explanations.

7. The structured and structured-adaptive tasks seem to have some potential problems when it comes to assessing their impact on mood ratings:i. the valence of the blocks was not randomised, meaning that the results could be confounded. E.g. what if negative RPE effects are longer-lasting than positive RPE effects? This seems plausible given the downward trend in mood in the random environment despite average RPE of zero. Could this also explain the pattern of mood in the other two tasks, rather than primacy?ii. scaling: if there is a marginally decreasing relationship between cumulative RPE and mood (such that greater and greater RPEs are required to lift/decrease mood by the same amount), then this will resemble a primacy effect? This is unlikely to be an issue in the random task but could be a problem in the structured and certainly in the structured-adaptive tasks.iii. individual differences in responsiveness to RPE: in the structured-adaptive task, some subjects' mood ratings may be very sensitive to RPE, and others very insensitive. One might expect that given the control algorithm has a target mood, the former group would reach this target fairly soon and then have trials without any RPEs, while the latter group would not reach the target despite ever increasing RPEs. In both cases the Primacy model would presumably win, due to sensitivity to outcomes in the first half or insensitivity to bigger outcomes in the second half respectively? Can the authors exclude these possibilities?

The reviewers raise several important concerns related to potential experimental confounds, that might have led to the Primacy weighting advantage. We would like to point out a few points that we believe address these concerns (and we also discuss these points in the manuscript on page 21):

First, we agree that a lack of randomization of block order is a weakness of the experiments we present, and we have added language to the discussion highlighting this point. However, we do not find evidence in our data for negative RPE effects being longer-lasting than positive ones. We find a sharp increase of mood at the beginning of the third block (from negative mood back up to positive mood) in the adaptive task, for example. Moreover, since the negative block in the adaptive task is second, negative RPEs being longer-lasting would on the contrary, interfere with a Primacy weighting of events, rather than explain it.

Second, as the reviewers mention, we find a primacy weighting of events on mood in the random task, as well as in the Structure task. Since RPEs do not adaptively increase over time in these tasks, it is unlikely that this is an explanation for the superior performance of the Primacy model in general. However, the difference between the primacy and recency models is larger in the Structured and Structured Adaptive tasks, and some of this difference may be due to an adaptation to rewards.

Third, we agree that the interaction between individual behavior and the controller in the Structured Adaptive environment could raise numerous interpretative difficulties. However, the Primacy model also fits better in the structured and random tasks, where the tasks did not respond to individual differences in responsiveness to RPEs. This indicates that the better fit of the Primacy model is unlikely to be solely driven by the adaptive controller in the Structured Adaptive task. However, it is possible that the controller does provide an advantage to the Primacy model over the Recency models in the Structured Adaptive task, and we have added these caveats to the discussion, as follows:

On page 21 of the Discussion:

“It is conceivable that what appears to be a primacy effect, is actually due to longer lasting effects of, say, positive RPEs on mood—this could be particularly exacerbated in the structured adaptive task. However, since our result was robust also in a random task design, as well as when testing a model with a varying time window parameter that considered different number of previous trials (t_max_, see Figure S3), we do not find evidence for the block valence order to account for the better performance of the Primacy model. In addition, the interaction between individual behavior and the controller in the structured-adaptive environment could raise interpretative difficulties. Therefore, it is important to stress that the Primacy model also fit better in the structured and random tasks, where the tasks did not respond to individual differences in responsiveness to RPEs. Yet, it is possible that this is a contribution to the fact that the advantage of the Primacy model over the Recency models is greater in the structured-adaptive task. There may also be additional mechanisms at play in the structured-adaptive task such as hedonic extinction towards RPEs that explain some of the increased performance of the Primacy model compared to the Recency model in this task.”

8. In relation to the fMRI analyses, the results in the main text seem to result from a second-level ANCOVA, where the individual weights of the Primacy model are shown to correlate with activation in the prefrontal cortex. Similar analyses using the weights of Recency model do not produce significant results at the chosen threshold. This analysis is problematic for two reasons. First, absence of evidence does not imply evidence of absence - was a formal comparison of the regression coefficients conducted? Second, to really validate the model the authors should show that the trial-by-trial correlates of expectations and prediction errors are more consistent with the Primacy than the Recency model, using a parametric analysis at the participant level.

To address this point, we now explain the comparison of the regression coefficients we have conducted in more detail. In this analysis, we contrasted between the regression coefficients of the two models and showed that the activation in the prefrontal cortex was more strongly correlated to the individual expectation weights of the Primacy model versus the Recency model (with t = 5.00, p = 0.0017, peak at [11,49,9], and a cluster size of 529 voxels). We explain this analysis in more details in the figure legend, as follows:

On page 19 of the Results, Figure 4:

“A formal comparison between the relation of brain activation to the Primacy versus the Recency models, was conducted. We compared the regression coefficients of the correlation between participants’ brain activation and the Primacy expectation term weights, versus the regression coefficients of the relation to the Recency model expectation term (see Figure S5 for the two images before thresholding and before contrasting against each other)”

Although we strongly agree that a trial level correlation would be an interesting analysis, we believe the number of trials in our task is insufficient to allow us to test this question with sufficient rigor. We definitely agree that this question of the trial-level correlations will be important to study, as it might inform of a different resolution of such task-brain relations, e.g., temporal processes that occur across trials and within blocks. Although this is of great interest to us, it is unfortunately beyond the scope of our collected data. However, we believe that our current results answer a different but not less important question, namely the relation between the overall individual weight of the expectation term across the entire task and the overall neural activation along the 34 mood ratings.

9. Similar to point 6, it is hard to conclude much about the models from the fact that the Primacy model E beta (but not the Recency model E beta) correlates with BOLD responses in a prefrontal cluster, when the Recency model E term is based on previous expectations, not previous outcomes. Likewise with the direct comparison of the models' voxel-wise correlation images.

We thank the reviewers for pointing out that we were unclear in our explanation of this. We hope that our updated explanation (legend of Figure 4) makes the point clearer. Both 𝛽*_E_* coefficients of the Recency and the Primacy models are based on previous expectations E, while the E term differs between the models in being either based on the current gamble options (in the Recency model) or the average of all previous outcomes (in the Primacy model). As our result shows (figure 4), the expectation term weight 𝛽_*E*_ from the Primacy model correlates significantly with bold responses in the prefrontal cluster (a peak at [-3,52,6], size of 132 voxels, peak beta = 44.80, t = 3.37, threshold at p = 0.0017). We then directly tested this correlation against the correlation of the 𝛽*_E_* of the Recency model, and found that the Primacy model 𝛽_*E*_ was more strongly correlated to brain activity (peak at [-11,49,9], t = 5.00, extending to a cluster of 529 voxels, threshold at p = 0.0017). This result provides a possible neural underpinning specific to the Primacy model’s mathematical realization of expectations and mood.

[Editors' note: further revisions were suggested prior to acceptance, as described below.]

Essential Revisions:1. The new Figure S1 is helpful, but caused some confusion as the description in the legend could be clearer. The reviewers assume that the blue bars are the weights attached to all outcomes (not expectations) from trial 1-8, all contributing to the expectation on trial 8; the orange bars are the weights attached to all outcomes from trial 1-7, all contributing to the expectation on trial 7; and so on. Assuming this understanding is correct, please amend the text in the legend to clarify this better, and also provide a colour key for the figure to make it clear that the bars refer to outcomes from specific trials.

Thank you for asking for this clarification helped us to improve the clarity of this figure to the readers. We have modified the text of the legend in the following way (on page 46-47):

“The Primacy effect of outcomes on mood. In the Primacy model, the expectation term Ej is the unweighted average of previous outcomes. At each trial, all previous expectation terms are combined in an exponentially weighted sum: ∑j=1tγt−jEj. Here we illustrate how this gives rise to a primacy weighting of previous outcomes that depends on the value of γ (each subplot represents a different magnitude of exponential weighting γ). The total height of each bar represents the influence of the outcome of the corresponding trial on the result of the exponential sum at the end of trial 9. Each color indicates the contributions of the outcomes that form an expectation term Ej at the end of trial *j*. The dark yellow block represents the contribution of the expectation term E1 from the end of trial 1 (comprised only of the first outcome). The grey blocks represent the contributions of the expectation term E2 that is being added from the end of trial 2 (which is the average of the outcomes from trials 1 and 2 and therefore it appears in both the first and the second bars). This continues for the rest of the expectation terms until the last expectation term E9 is added, which is formed by averaging the outcomes from trials 1 to 9 as shown by the blue bars. “

2. The Primacy vs Recency model comparison is of critical importance, so it would help the paper to be as clear as possible about it. The key comparison here is between the variant of the Recency model that is identical to the Primacy model in terms of having both a learned outcome as the expectation and without the certainty term. Your model comparison seems to test these changes separately rather than together (apologies if we have misunderstood this).Can you explain what it is, specifically, about the Primacy model that causes it to perform better than the best Recency one (e.g. is it the nature of the way the expectation is learned, or something else)? As we understand it, the central claim of your paper is that people learn the expectation term in a way that effectively puts more weight on the initial outcomes encountered, so it would be really useful to understand what it is about the learning process that causes this effect, and whether it is this that improves the fit of the model.

We have addressed this modelling question by building an additional model as suggested, that has both a learned dynamic outcome probability as expectation and no Certain value term. We have tested for these two properties separately before to follow a simple one modification at a time process, but we greatly agree with the importance of testing such a merged Recency model that is most similar to the Primacy model. We have therefore run all the analyses of this manuscript again with this new merged model (termed the Recency with both dynamic win and no Certain term model). We now present in the paper these new results which strengthen and support the conclusion of this paper, as we found that the Primacy model also preforms better than this more similar Recency model.

Our conclusion that the effect of outcomes on mood through expectations has a primacy weighting in our tasks holds robustly when we consider a variety of different but similar models which either have primacy weighting (Figure 3-figure supplement 2) or recency weighting (Table 1). All the models with primacy weighting share that the expectation is based on an average over previous outcomes or potential outcome values. We stress that the expectation itself does not have to have primacy weighting for our conclusions to hold. The primacy model that we have chosen as our representative primacy model (due to having superior or statistically indistinguishable performance over the alternative primacy models) applies equal weights to all past outcomes to form the expectation, but we have also tried models where the weighting within the expectation had higher weights for more recent outcomes. In all these cases, the combination of current and past expectation still results in a primacy weighted aggregate effect of previous outcomes on mood. The dependence of mood on an accumulation of previous expectations is therefore what causes the primacy weighting, as the initial outcomes have a larger influence on mood versus a smaller influence of past expectation terms. In an intuitive sense, the primacy effect represents the greater weight first experiences have in a new environment or context, simply by virtue of coming first. The first event has nothing against which it can be compared, the second event has only itself and the first; the third event can be compared only against the first two, and so on, till eventually each additional event has a minimal impact in the face of all the events that have come before. The more trials we experience, the more information we gain, and the less meaning each event has on its own. This process has clear parallels to learning, but our models are agnostic to the exact mechanism by which expectations are accumulated. It is likely that there are equivalent formulations to our models in which expectation is a learned parameter controlled by a learning rate. The details of this mechanism are certainly of interest, but these will need to elucidated by future studies.

We have added the above interpretation of the primacy model to the Discussion on pages 26-27.

3. Relating to the above point, the key comparisons between Primacy and Recency models in the main manuscript (e.g. the comparative fMRI analyses) should be between these similar models, not the Primacy and original Recency model.

Method:

As requested, we have performed all the analyses throughout this manuscript using the new merged Recency model which is most similar to the Primacy model, termed the “Recency with both dynamic win and no Certain”. These analyses included model comparison on fitting the data from all the different datasets, i.e., the random, structured, and structured-adaptive tasks, adult and adolescent samples, clinically depressed adolescents, online and lab-based conditions. Moreover, the new model coefficients were used to conduct again the brain activity correlation analysis, to search for subject-level correlations between this merged Recency model variant and neural activity when rating mood.

Results:

The new Recency model with both dynamic win probability and no Certain term performed better on all the different datasets and samples used in this study, according to the streaming prediction criterion (see the new Table 1 that presents these values and the new section of the Results that summarizes this work).

Moreover, brain analysis revealed a similar result to the results with the original Recency model. No significant neural correlations were found to this model’s coefficients, but only to the primacy model coefficients (there were also no frontal clusters before thresholding the images, as shown in the Supplement figure to Figure 4 where the non-thresholded images of this analysis are presented). We have conducted both the whole-brain level analysis with the new merged Recency model and also contrasted these resulting images against the resulting images of the Primacy model, to formally show the stronger relation of the Primacy model against this new Recency model as well (see legend of Figure 4 for these results).

Changes made:

First, the ordering of the manuscript was changed to match the presentation of the merged Recency model now throughout the sections of the results (thus presenting the different Recency models prior to reporting the results).

Results section on page 10:

“Next, we merged the dynamic win probability and the elimination of the Certain term to an additional Recency model that is most similar in its characteristic to the Primacy model (i.e., the “Recency with both dynamic win and no Certain model”). “

Table 1 on page 16: Now reports the performance of the Primacy model against the new Recency model as well.

Results section on page 18:

“We correlated BOLD signal with the participant-level weights of the parameters of the Primacy and two of the Recency models (the original Recency model and the Recency model that is most similar to the Primacy model, i.e., the one with both dynamic win probability and no Certain term).”

Figure 4 legend, on pages 19-20:

“(see Figure 4-figure supplement 1 for the images of the relation to the Primacy model and each of the Recency models before thresholding and before contrasting against each other). This contrast showed a significantly stronger relation of the Primacy model expectation weight to brain signals around the ACC region (for p = 0.0017), with 529 voxels around [-11,49,9] for the original Recency model and 328 voxels around [-11.2, 48.8, 3.8] for the Recency with both dynamic win and no Certain term model”